

# AI-Driven TanDEM-X Penetration Bias Estimation in Antarctica Using ICESat-2 and ECMWF Data: Implications for the NASA Surface Topography and Vegetation Decadal Survey Incubation study

Ankita Vashishtha[1], Pietro Milillo[1,2,3], Alexandre Becker Campos[3,5], Jose Luis Bueso Bello[3]
, Paola Rizzoli[3], Johan Nilsson[4]

[1]Department of Civil and Environmental Engineering, University of Houston, Houston, TX, USA
[2]Department of Earth and Atmospheric Science, University of Houston, Houston, TX, USA
[3]German Aerospace Centre (DLR), Microwaves and Radar Institute, Munich, Germany
[4]Jet Propulsion Laboratory, California Institute of Technology, Pasadena, CA, USA
[5]FAU Erlangen- Nürnberg, Institute of Geography, Erlangen, Germany

*Correspondence to*: Pietro Milillo (pmilillo@cougarnet.uh.edu)

## Abstract

TanDEM-X radar penetration into polar ice introduces significant uncertainties in digital elevation models (DEMs),
consequently affecting the accuracy of glacial mass balance estimations. This limitation arises from the variable penetration depth of radar signals in snow and ice, which can cause the DEM surface to deviate from the true physical surface. X-band radars, operating in the 8–12 GHz range, interact with ice through absorption, reflection, and transmission. These interactions heavily depend on variables such as ice temperature, water content, snow density, and salinity, which directly influence microwave penetration depth and signal behaviour. Advancements in remote sensing have leveraged X-band synthetic aperture
radar (SAR) to characterize snow and ice surfaces, facilitating applications such as sea ice classification and snow layer analysis. However, X-band SAR signals often exhibit biases in elevation measurements due to their partial penetration into snow and ice. This bias complicates efforts to integrate SAR-derived digital elevation models (DEMs) with laser altimetry data like ICESat-2, where penetration is negligible. Therefore, correcting X-band biases is critical for achieving high-accuracy surface elevation models and improving glaciological interpretations. Here we address these challenges by integrating neural
network techniques with TanDEM-X (TDX) DEMs, Ice, Cloud, and land Elevation Satellite-2 (ICESat-2) altimetry data, and environmental parameters from European Centre for Medium-Range Weather Forecasts (ECMWF). We leverage about 300,000 ICESat-2 pointwise measurements acquired within 30 days of TDX measurements spanning 2021-2024 in Antarctica. Additionally, we consider a diverse dataset of snow and atmospheric variables including temperature, snowfall, snow depth, and wind speed to model and predict X-band penetration biases across Antarctica. This approach advances existing methods
by automating bias correction and enhancing the integration of SAR and altimetry datasets achieving mean bias correction of the order 1 cm with a Root Mean Square Error (RMSE) of about 1 m and maximum errors of the order of 10 m. Findings from this work provide actionable insights for improving elevation model accuracy in different ways. First, we offer the retrieved



pointwise TanDEM-X, ICESat-2 and ECMWF dataset open access for future studies through a dedicated Github page. Second, we provide both our trained network weights and our algorithms in the form of Jupyter Notebooks for improved reproducibility.

Most importantly, we discuss the broader efforts in the NASA Surface Topography Elevation (STV) decadal survey incubation study and polar ice monitoring by addressing penetration biases which is one of the most critical uncertainties in radar remote sensing of snow and ice.

## 1 Introduction

Accurately characterizing ice sheet surface elevation and mass balance in Antarctica is fundamental to understanding polar climate dynamics and sea-level rise (Pritchard et al., 2009;Flament & Rémy, 2012; Paolo et al., 2023; Fricker et al., 2025). Covering an area of 14.2 million square kilometers (5.5 million square miles), remote sensing techniques provide the most effective means of measuring ice sheet mass balance and its variations. A range of remote sensing methods are available, including laser altimetry with ICESat-1 and ICESat-2 (Sørensen et al., 2011;Leigh et al., 2015; B. E. Smith et al., 2023), radar

altimetry with CryoSat-1 and CryoSat-2 (Nilsson et al., 2016; Slater et al., 2018; Sørensen et al., 2018;Jakob et al., 2021), stereo-photogrammetry (Berthier & Brun, 2019;Howat et al., 2019;Dehecq et al., 2020), gravimetry using GRACE-1 and GRACE Follow-On (Groh et al., 2014;Tapley et al., 2019; Sasgen et al., 2020; Velicogna et al., 2020) and synthetic aperture radar bistatic interferometry with TanDEM-X (Rott et al., 2021;Abdullahi et al., 2024;Bannwart et al., 2024;Deng et al., 2024). Combining these remote sensing techniques allows for a comprehensive monitoring of ice sheet dynamics, improving our

understanding of ice mass loss, accumulation rates, and the impact of climate change. Each technique is characterized by advantages and drawbacks. For example, laser altimetry (ICESat-1 and ICESat-2) provides high-precision elevation measurements but is limited by cloud cover and requires repeated observations to capture temporal variations (Markus et al., 2017 ;Michaelides et al., 2021). Radar altimetry (CryoSat-1 and CryoSat-2) can penetrate clouds and operate in all weather conditions but may introduce biases due to surface roughness and radar penetration into snow (Michel et al., 2014; Dawson &

Landy, 2023). Stereo-photogrammetry offers high spatial resolution but relies on clear-sky optical imagery and solar illumination, making it susceptible to atmospheric disturbances (Bamber & Rivera, 2007;Racoviteanu et al., 2008) . Gravimetry (GRACE-1 and GRACE Follow-On) effectively detects large-scale mass changes but lacks fine spatial resolution, limiting its ability to resolve localized ice mass variations (Tapley et al., 2019; Velicogna et al., 2020).

Here we focus on X-band (3 cm wavelength) bistatic SAR Interferometry (InSAR) from the German TanDEM-X

mission (Milillo et al., 2019; Milillo et al., 2022) and NASA ICESat-2 altimetry measurements.  Specifically, we have extracted more than 300,000 data points of ICESat-2 acquisition acquired within 30 days of TanDEM-X. Our dataset includes the grounding line regions of seven glaciers located in East Antarctica including Amery and Cook Ice Shelves, Denman, Reid, Totten, Moscow University, and Holmes Glaciers, and additionally covers Jutulstraumen Glacier in Queen Maud Land, and Stancomb-Wills Glacier in the Weddell Sea (Figure 1). For these data points, we have retrieved TanDEM-X DEM height and

corresponding penetration bias values along with other features including amplitude, coherence, local incidence angle, height of ambiguity, and surface slope. Further to investigate the effect of physical parameters on penetration depth, we have derived,





14 environmental features listed in Table 1 extracted from the ECMWF global climate ERA-5 dataset with a horizontal resolution of 0.25 arc degrees (H. Chen et al., 2023). Additionally, to account for acquisition time parameters, all datasets were selected at monthly intervals from April to June during the years 2021-2023, which pertains to the winter season in the

Antarctic region. We then implemented five machine learning and nine deep learning algorithms, leveraging radar and environmental features to predict penetration bias values. A comprehensive analysis was subsequently conducted to evaluate the influence of each feature on the accuracy of bias estimation.

       The paper is structured as follows: Section 1 provides an overview of the physical theory underlying radar penetration biases and reviews relevant literature. Section 2 details the study areas and datasets used in the analysis. Section 3 outlines the

methodology employed, while Section 4 presents the results of our analyses. Section 5 discusses the significance of these findings, particularly in the context of NASA's Surface Topography and Vegetation (STV) decadal survey incubation study and future cryosphere research. Finally, Section 6 summarizes the main conclusions drawn from this study.

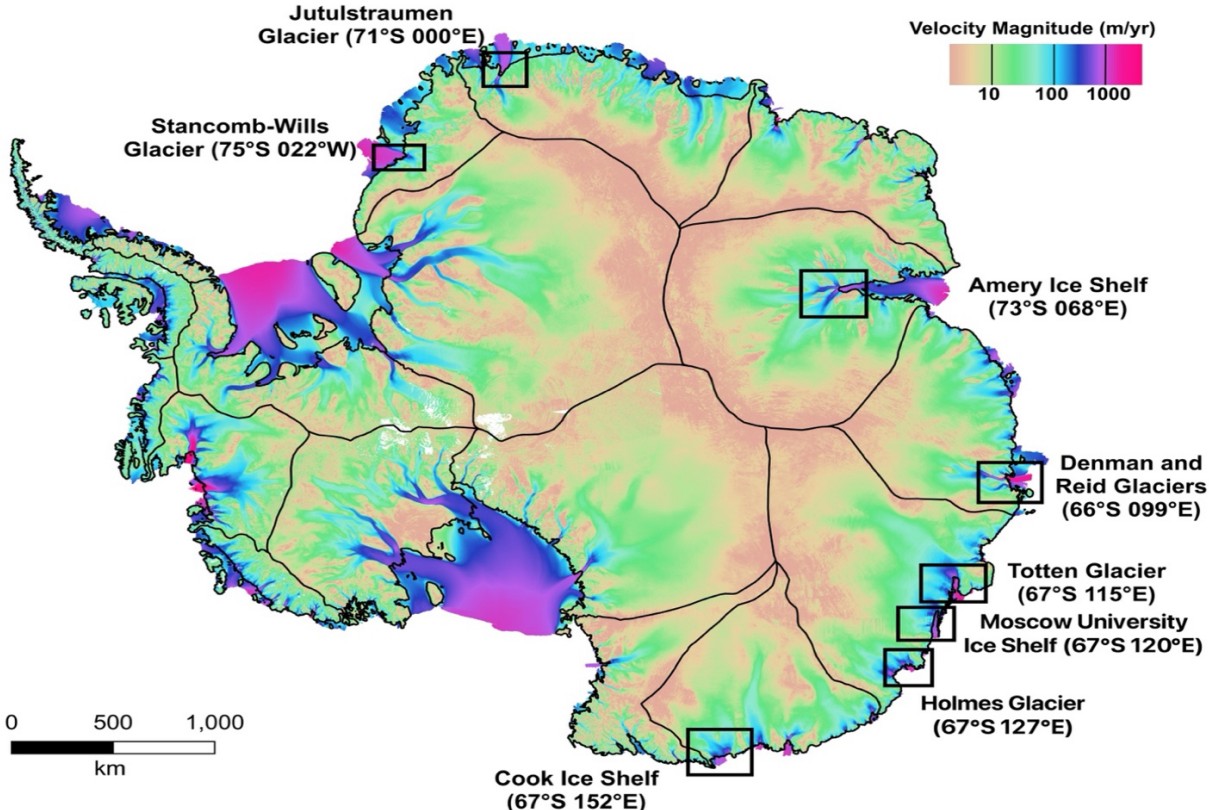

**Figure 1. Antarctica map showing the location of the regions in Antarctica for which ICESat-2 data, TanDEM-X data**
**and ECMWF features is extracted. Black rectangles show the name along with latitude and longitude coordinates of the eight regions. Velocity magnitude from 1996 to 2016 (Mouginot et al., 2017) is shown in logarithmic scale. "S" indicates South, and "E" indicates East direction. Antarctica IMBIE basin boundary is also shown on the velocity map (Rignot E. et al., 2011).**



**Table 1. Details of ICESat-2 elevation values, radar features extracted from raster files and environmental features extracted from ECMWF Reanalysis dataset.**

| Feature | Description and Source | Unit | Spatial Resolution |
|---|---|---|---|
| ICESat-2 Elevation data | ICESat-2 elevation data point from ICESat-2 data | m | 20 m along orbital track |
| TanDEM-X DEM height | TanDEM-X DEM height from TanDEM-X DEM raster files | m | $2 \times 10^{-4}$ arc degrees or 12m |
| Coherence | SAR bistatic coherence | NA | $2 \times 10^{-4}$ arc degrees or 12m |
| Amplitude | SAR backscattered signal $\beta^0$ coefficient | Decibel | $2 \times 10^{-4}$ arc degrees or 12m |
| Height of Ambiguity | Height difference that results in a $2\pi$ interferometric phase change after interferogram flattening | m | $2 \times 10^{-4}$ arc degrees or 12m |
| Local Incidence Angle | Angle between the incoming radar beam and the local normal to the surface at the point where the radar signal interacts with the terrain generated from TanDEM-X raw DEM raster files | Radians | $2 \times 10^{-4}$ arc degrees or 12m |
| Slope | Slope values calculated from TanDEM-X DEM raster files using the GDAL DEM slope function (OSGEO 2025) in python | degrees | 12m |
| Penetration Bias | Calculated as difference between TanDEM-X elevation and ICESat-2 elevation | m | NA |
| 2m temperature | Temperature of the air at 2m above the land surface from ECMWF | Kelvin (K) | 0.25 arc degrees |
| Snowfall | Accumulated snow that falls to the Earth's surface from ECMWF | m of water equivalent | 0.25 arc degrees |
| Surface Pressure | Atmospheric pressure at the land surface, sea and inland water from ECMWF | Pa | 0.25 arc degrees |
| Near IR albedo for direct radiation | Fraction of direct solar radiation in Infrared (IR) range of 0.7 to 4 μm reflected by the Earth's surface from ECMWF | NA | 0.25 arc degrees |
| Snow Density | Snow density per cubic meter in the snow layer from ECMWF | kg/m$^3$ | 0.25 arc degrees |
| Snow Depth | Amount of snow from the snow-covered area of a grid box from ECMEF | m of water equivalent | 0.25 arc degrees |
| Snow Evaporation | ECMWF Accumulated amount of water that has evaporated from the snow-covered area into the atmosphere as vapor | m of water equivalent | 0.25 arc degrees |
| Temperature of snow layer | Temperature of the snow layer from the ground to the snow-air interface from ECMWF | Kelvin (K) | 0.25 arc degrees |
| Total column snow water | Total amount of water in the form of snow (includes collective ice crystals which can fall to the surface as precipitation) in a column | kg/m$^2$ | 0.25 arc degrees |



| | extending from the earth surface to the top of the atmosphere from ECMWF | | |
|---|---|---|---|
| Total Precipitation | Accumulated liquid and frozen water, consisting of rain and snow that falls to the Earth's surface. It is an aggregation of large-scale precipitation and convective precipitation from ECMWF | m | 0.25 arc degrees |
| Wind speed | Magnitude of eastward (10m u-component of wind) and northward component (10m v-component) of the wind at a height of ten metres above the Earth surface from ECMWF | m/s | 0.25 arc degrees |
| Wind direction | Angle between the eastward (10m u-component of wind) and northward component (10m v-component) of the wind at a height of ten metres above the Earth surface from ECMWF | Degree | 0.25 arc degrees |
| Snow Albedo | Measure of the reflectivity of the snow-covered part of the grid box. It is the fraction of solar radiation reflected by snow across the solar spectrum from ECMWF | NA | 0.25 arc degrees |
| UV visible albedo for direct radiation | Fraction of direct solar radiation with wavelength varying between 0.3 to 0.7 μm reflected by the Earth's surface from ECMWF | NA | 0.25 arc degrees |

## 1.1 Penetration of X-band microwaves in ice and snow

X-band signals interact with the ice surface and subsurface through absorption, reflection, and transmission. The depth to which these signals penetrate depends on snow and ice properties, which are further modulated by meteorological and climatic factors (Figure 2). Previous research indicates that the penetration depth of X-band microwaves diminishes with increasing snow thickness(Nandan et al., 2017). This attenuation is influenced by the dielectric properties of snow, which are determined by factors such as density and liquid water content. Elevated snow density and moisture levels typically lead to a reduction in penetration depth (Page & Ramseier, 1975; Guo et al., 2023). Additionally, the dielectric constant of ice is affected by temperature and salinity; lower temperatures and higher salinity contribute to decreased penetration depth (Achammer & Denoth, 1994). As such X-band is more sensitive to changes in snow salinity in comparison with other microwave bands (Nandan et al., 2017).

InSAR coherence also effects the penetration depth showing deeper penetration in low coherence region due to enhanced volume scattering(Deng et al., 2024). Most importantly, X-band penetration varies significantly between dry and wet snow due to their different dielectric properties. In dry snow, X-band microwaves penetrate more deeply because dry snow primarily consists of air and ice, which have lower absorption rates. This allows for larger backscatter returns from the underlying ground due to deeper scattering (Leinss et al., 2014; Park et al., 2021). Wet snow in contrast contains liquid water, which significantly increases absorption and reduces penetration depth. As a result, backscatter is primarily from the air-snow interface, and overall backscatter intensity decreases due to surface scattering ( Park et al., 2021) . Increase in snow depth and snow roughness considerably affects the microwave backscatter by scattering radiation at the surface and restricting its



penetration in the subsurface of the snow layers (Park et al., 2021). Overall, to describe the interaction of X-band signal with surface snow, we introduce the concepts of penetration depth and penetration (or elevation) bias (Rott et al., 2021).Penetration

depth refers to the distance an electromagnetic signal travels through snow before being significantly attenuated, whereas penetration bias describes the elevation measurement error arising when radar signals reflect from subsurface layers rather than from the actual snow surface. (McGrath et al., 2019;Abdullahi et al., 2024;Bannwart et al., 2024).

**1.2 TanDEM-X penetration in snow and ice**

120          In TanDEM-X bistatic InSAR observations, the location of the phase center and in turn penetration bias is influenced by ice physical parameters like snow volume, snow density, surface roughness, water content, absorption, and scattering losses (Abdullahi et al., 2024). Additionally, penetration bias is equally influenced by sensor properties of frequency, polarization or acquisition geometry. Seasonal variability also effects the scattering patterns of a target surface. However, penetration bias is primarily governed by the electrical property known as relative permittivity or dielectric constant (Page & Ramseier, 1975).

The dielectric constant of snow is directly or indirectly influenced by the physical parameters described previously, thereby significantly affecting the penetration bias of X-band radar signals. Both penetration depth and bias can be related to InSAR bistatic coherence, which is a measure for the accuracy of determination of interferometric phase. Bistatic SAR Coherence ($\gamma$) is inversely related to the two-way penetration depth ($\delta_2$) with the elevation bias ($\Delta h$)caused by signal penetration bounded by this relation. This bound on elevation bias depends on both coherence ($\gamma$) and the height of ambiguity ($h_a$). In regions of

low coherence, the penetration depth and bias tend to be higher, whereas high coherence regions show less penetration depth and minimal bias values.

          The ratio of penetration bias and height of ambiguity $\Delta h/h_a$ is equivalent to the relative penetration depth $\delta_2/h_a$ (Dall, 2007). Height of ambiguity ($h_a$) is the height difference that generates and interferometric phase change of $2\pi$ after interferogram flattening(Dall, 2007). For an infinitely deep volume scatterer surface having negligible surface scattering, two-

way penetration depth and bias are almost equals if the penetration depth is much smaller than the height of ambiguity. However, when penetration depth exceeds 10% of the height of ambiguity, the penetration bias becomes 10% less than the penetration depth (Dall, 2007).

**1.3 Penetration Bias Estimation in this work**

          In this study we are correcting for elevation biases therefore ignoring the refraction that occurs as X-band travels

through the glacier ice. In other words, we have calculated the penetration bias as the elevation difference between TanDEM-X elevation and ICESat-2 elevation and hence not considering the repeated refraction of the X-band passing through the ice medium. Penetration depth is more complex phenomenon which occurs due to X-band signal attenuation as it transmits through the ice medium and hence considers the refraction for every change in refractive index of the ice layers and hence considers the transmission angle. Extensive literature has examined TanDEM-X penetration biases in Antarctica, revealing its

dependence on snow and ice properties. TanDEM-X measurements indicate that in dry snow areas, X-band SAR signals penetrate to depths of approximately 10–12 meters (Rizzoli, et al., 2017). The penetration depth also varies with snow



conditions, as observed in Greenland, where it averages 3.7 meters in wet snow zones and increases to 5.4 meters in dry snow regions(Park et al., 2021). Blue ice areas (BIAs) in Antarctica serve as important reference sites for validating X-band InSAR measurements since their high ice density prevents signal penetration, enabling precise elevation comparisons with laser

altimetry data (Rott et al., 2021; Wessel et al., 2021). To account for penetration biases, researchers have developed correction methods that integrate InSAR coherence, backscatter intensity, and comparisons with laser altimetry (Hoen & Zebker, 2000; Stebler et al., 2005;Dall, 2007; Oveisgharan & Zebker, 2007; Michel et al., 2014;Abdullahi et al., 2019 ; Fischer et al., 2019; Fischer et al., 2020; Li et al., 2021;Park et al., 2021;Rott et al., 2021; Abdullahi et al., 2024;Bannwart et al., 2024) Given that X-band radar can penetrate significantly into glaciers, particularly in dry snow conditions, these findings have critical

implications for elevation change assessments and mass balance studies in Antarctica.

       In previous studies, penetration bias is evaluated by various data-driven models by interpreting the scattering mechanism, which depends on the physical properties of surface snow (Hoen & Zebker, 2000; Stebler et al., 2005; Dall, 2007; Oveisgharan & Zebker, 2007; Abdullahi et al., 2019 ; Fischer et al., 2019; Park et al., 2021). Models like the coherent backscatter method (Dall, 2007; Abdullahi et al., 2019; Fischer et al., 2020) are calibrated by applying penetration biases

inferred through bistatic coherence loss. Further penetration biases are also estimated by conducting an empirical analysis of facies zones as these features reflect specific scattering attributes of the snow in that region and hence can be utilized for the approximation of bias (Fahnestock et al., 1993;Rignot et al., 2001; Falk et al., 2015; Gray et al., 2015) Additionally, fuzzy classification as an effective approach is also applied to demarcate facies for further bias estimation (Rizzoli et al., 2017). All the previous modelling approaches estimated the penetration bias by relating with a few variables like coherence, SAR

backscatter intensity with underlying assumptions or by equating some environmental parameters like snow water equivalent, snow density, snow liquid water content and top layer snow temperature to the backscatter intensity (Park et al., 2021).

       Prior models were applied either to image datasets or to a small region where altimetry data set was available for bias calculation. However, penetration bias is significantly influenced not only by the sensor variables like frequency, polarization, and viewing geometry but also by the physical snow properties (Park et al., 2021). The novelty of this work lies in the

evaluation of penetration biases using non-linear regression methods implemented through various machine learning and neural network (NN) approaches. Study areas and dataset used are described in the next section of this paper.



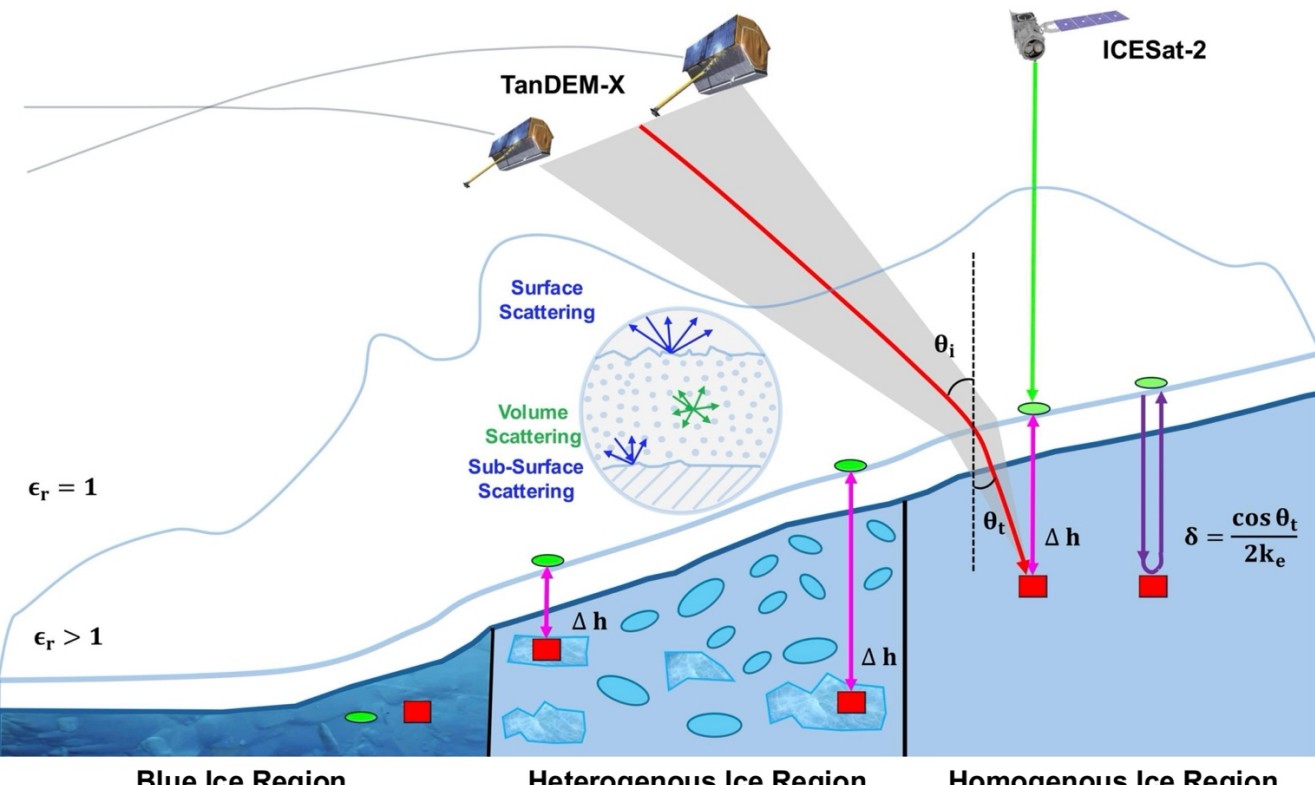

**Figure 2. X-band microwave penetration in the Homogeneous, Heterogenous and Blue Ice region on a glacier demarcated clearly with different patterns in the figure. $\theta_i$ is the local incidence angle, $\theta_t$ is transmission angle and $\Delta h$ is the penetration bias for TanDEM-X InSAR acquisition. $\delta$ is the 2-way penetration depth which depends on $\theta_t$ transmission angle and $k_e$ extinction coefficient of the ice medium. ICESat-2 footprint is indicated by green circles and location of InSAR phase centre is represented by red rectangle.**

## 2 Study Area and Data

We use over 300,000 data points from ICESat-2 acquisitions for eight different grounding zone regions in East Antarctica including Amery and Cook Ice Shelves, Denman, Reid, Totten, Moscow University, and Holmes Glaciers, and additionally cover Jutulstraumen Glacier in Queen Maud Land, and Stancomb-Wills Glacier in the Weddell Sea (Figure 1). These data have been collected between April and June, the typical winter season, during the years 2021 to 2024. Apart from ICESat-2 elevations, other radar features namely TanDEM-X DEM height, coherence, local incidence angle, height of ambiguity, amplitude, slope are also added for all the data points. Bias values are calculated as difference between ICESat-2 and TDX elevation values. The time gap between ICESat-2 and TanDEM-X elevation data ranges within ± 30 days. When analyzing penetration bias between TanDEM-X (TDX) and ICESat-2 altimeters, the time interval between acquisitions is a critical factor, especially for surface types that may undergo change (e.g., snow accumulation, melt, or densification). There's no strict




maximum allowable interval, but studies suggest the time difference should ideally be within 1–2 years to ensure surface comparability and minimize bias caused by temporal evolution. In (Abdullahi et al., 2019) a detailed assessment over the Greenland Ice Sheet found that TDX (2009–2015 acquisitions) could be compared to ICESat-2 (post-2018) data, but only in regions where surface elevation changes were assumed to be negligible or corrected with model data. This suggests that up to 3–5 years can be acceptable if surface change is well modelled or stable. (MacGregor et al., 2021) highlight the value of multi-year datasets like Operation Ice Bridge to bridge gaps and validate penetration correction methods across sensors. However, in high-accumulation or dynamic zones (e.g., coastal Antarctica or glaciers), shorter time gaps, ideally less than a year, are preferred to minimize vertical discrepancies due to surface evolution. For these reasons we select a maximum time difference between TDX and ICESat-2 acquisitions less or equal to 30 days. Further, 14 environmental features are also added from ECMWF ERA-5, available at hourly basis, at a horizontal resolution of 0.25 arc degrees are added for all these data points. Details of all these features are given in the Table 1.

## 2.1 ICESat-2 data

The ICESat-2 launched in 2018, was a follow-up mission of ICESat which was operational from 2003 to 2009(Zwally et al., 2002). It consists of an Advance Topographic Laser Altimeter System (ATLAS) that transmits a laser beam with a wavelength of 532nm at a pulse repetition frequency of 10kHz (Markus et al., 2017). ICESat-2 operates with three pairs of laser beams with each pair is separated by distance of 3km in cross-track with a pair spacing of 90 m which enables ICESat-2 to determine the local cross track slope. The nominal diameter of footprint of each of the beams is 17m with an along track sampling interval of 0.7 m due to high repetition rate of 10 kHz (Markus et al., 2017). For Land Ice measurement, the ATL06 dataset (B. Smith et al., 2019)is utilized having a resolution of 20 m along track (using a +/- 20 window) and an elevation accuracy of better than 3cm (Brunt et al., 2019).To retrieve elevation profiles, the collected dataset of point elevation measurements is sampled by utilizing different beam spacings and arbitrary repeat-track geometries, along with the estimation of Root Mean Square (RMS) errors result into the elevation change measurements. For interpolating the beam track to reference ground track (RGT), ICESat-2 requires to control the beam position to less than half the pair separation. The orbit of ICESat-2 is inclined at 92° along with measurement range up to 88° north and south, with a repeat cycle of 91 days (Markus et al., 2017). The elevation precision of the data depends on three factors namely signal-to noise ratio, distance at which laser footprints are collected and the precision in timing the photons. Each beam pair is consisting of strong and weak beam lasers which facilitates in the calculation of cross track slope, an important parameter in cryosphere research. The NASA cap toolkit (Paolo and Nilsson 2025) was used to produce and process the ICESat-2 data.

## 2.2 TanDEM-X data

TanDEM-X, launched on 21st June 2010, is an extension mission of TerraSAR-X, with both satellites moving synchronously in a close formation in a helix orbit relative to each other to mitigate collision risks at the poles(Zink et al., 2021). The instrument in both satellites consists of an advanced X-band SAR bistatic system capable of acquiring data in different modes



in full polarization with the centre frequency of the radar at 9.65 GHz and a chirp bandwidth up to 300 MHz (Krieger et al.,
2013). The two satellites are separated by a few hundred meters to enable data acquisition through single-pass and cross-track
interferometry or InSAR bistatic acquisitions. By analysing the phase difference between two coherent radar signals acquired
from slightly different spatial locations, TanDEM-X achieved metric precision in the measurement of range difference between
the satellites and scatterer on ground. TanDEM-X is also designed to maintain a sharp balance between height of ambiguity
with phase disambiguation(Zink et al., 2021).

This work utilized 413 TanDEM-X derived DEMs covering eight regions listed in (Table 2) (Figure 1). To enhance
signal coherence and consistency of DEMs, all SAR and ICESat-2 data are considered in the winter season month of April,
May and June from 2021 to 2024. The bistatic co-registered single-look complex (CoSSCs) SAR data of TerraSAR-X and
TanDEM-X is processed by the Integrated TanDEM-X Processor (ITP) of the German Aerospace Center (DLR). ITP is used
to produce pre-calibrated and geocoded single-scene raw DEMs nearer to ICESat-2 acquisition date over the East Antarctica
region. The ITP includes data take screening, focusing and interferometric processing steps including radargrammetry for
absolute range DEM estimation (Rossi et al., 2012). The interferometric processing generating geocoded DEMs has been
performed in the TAXI processing chain (Prats et al., 2010). In additions to the DEM, interferometric coherence, backscatter
intensity, height of ambiguity and local incidence angle raster images are also extracted to analyse the penetration bias. All
TanDEM-X datasets are bilinearly resampled to the spatial resolution of 12m with reference to WGS84, to align with the
independent pixel spacings of the interferometric processing filters and to minimize random height errors through averaging
(Martone et al., 2018). According to previous literature studies, the absolute vertical height accuracy and horizontal accuracy
of the standard TDX DEMs is better than 10 m with circular error and linear error at 90% confidence interval for all the
TanDEM-X datasets,(Zink et al., 2021; Gonzalez et al., 2024). Whereas the relative vertical height accuracy is smaller than
2m for low and medium relief terrain and 4m for high relief terrain on 1° × 1° grid cell with linear errors at 90% confidence
interval (Rizzoli, et al., 2017).


**Table 2. Details of raw DEM tiles for each of the eight regions in the Antarctic region**

| Antarctica Region | Name of the Location | Coordinates of the Location | No. of DEM tiles | DEM Years |
|---|---|---|---|---|
| East Antarctica | Cook Ice Shelf | 69°S 152°E | 55 | 2021-2023 |
| East Antarctica | Holmes glacier | 67°S 127°E | 28 | 2021-2023 |
| East Antarctica | Moscow University Ice Shelf | 67°S 120°E | 68 | 2021-2023 |
| East Antarctica | Totten glacier | 67°S 115°E | 90 | 2021-2024 |
| East Antarctica | Denman and Reid glaciers | 66°S 099°E | 53 | 2021-2023 |





| East Antarctica | Amery Ice Shelf | 73°S 068°E | 32 | 2021-2023 |
|---|---|---|---|---|
| Weddel Sea | Jutulstraumen glacier | 71°S 000°E | 39 | 2021-2023 |
| Queen Maud Land | Stancomb-Wills glacier | 75°S 022°W | 48 | 2021-2024 |

**2.3 ECMWF ERA5 data**

Fourteen atmospheric features complement our dataset (Table 1). These atmospheric features are extracted by ERA5 data
which is a fifth generation ECMWF reanalysis of the global climate and weather. These data are available from 1940 onwards
on hourly basis(H. Chen et al., 2023). Reanalysis data combines model data used by numerical weather prediction centres and
assimilate the updated data with previous forecast data to predict the optimal values of atmospheric features. The horizontal
resolution of this data is about 0.25 arc degrees for the reanalysis and 0.5 degrees for the uncertainty estimate. In this work,
ERA-5 reanalysis data for each atmospheric feature is downloaded for each TDX acquisition.


**3 Methodology**

The overall workflow of the methodology is shown in Figure 3. Input data is fed into the machine learning and deep learning
algorithms through Comma-Separated Values (CSV) file consisting of the value of all the radar and atmospheric features for
each ICESat-2 data point.







**Figure 3. Workflow of the methodology. KNN stands for K- nearest neighbour, GBM stands for gradient boost machine, XG stands for extreme boost, ResNet stands for residual network, NN stands for neural network, ReLU stands for rectified linear unit, MLP stands for multi-layer perceptron, RMSE stands for root mean square error, MAE stands for mean absolute error.**



### 3.1 Dataset Preparation

Input features is described in Table 1. Initial parameters of the CSV file consist of latitude and longitude of the ICESat-2 data points in WGS 84 EPSG 4326 map projection along with the elevation values from ICESat-2 data sets, datasets are converted to the WGS84 ellipsoid vertical datum. TanDEM-X data consist in raster tiles in WGS 84 EPSG 4326 map projection. For each ICESat-2 data point, the corresponding TanDEM-X elevation data and associated radar features is extracted from the raster acquired on the date closest to ICESat-2 observations, provided that the raster spatially overlaps with the data point

within the region of interest. TanDEM-X elevation data is extracted by applying a 3×3 moving window centred on the raster cell corresponding to each ICESat-2 data point. Slope raster files are generated by GDAL library in python for each individual DEM raster. The value for elevation, slope and other radar features is evaluated as the mean of all the raster cell values within this window. This averaging approach helps to mitigate random variations and small-scale fluctuations in the raster data, thereby enhancing the reliability of the extracted values for terrain analysis applications. Without any correction applied the

differences between ICESat-2 elevation data and TanDEM-X DEM height are characterized by an RMSE of 4 m and median and mean value of -1.6 and -1.48 m along with minimum, maximum values as -18 m, 20 m respectively in line with previous studies (Rizzoli, et al., 2017).

Firstly, we have removed the data points lying in the floating ice region by removing the data points having TanDEM-X DEM height less than 100m. We consider only points where the TanDEM-X coherence is larger than 0.3 as high coherence

values lead to a more accurate estimate of the interferometric phase. Similarly to (Gonzalez et al., 2024) we consider InSAR DEMs finite accuracy due to perpendicular baselines. These types of error appear as low-varying offsets and tilts (Krieger et al., 2013) . For heights of ambiguity ranging between 30 and 80 meters expected tilts in a scene range between 7 to 19 cm (Gonzalez et al., 2024). Additionally, points with a height of ambiguity less than 30 meters were excluded, as they constitute only less than 1% of the dataset and could introduce imbalance in the training set. Therefore, our models will be applied only

to InSAR data characterized by height of ambiguity comprised between 30 and 80 meters. We also filter out points characterized by slopes steeper than 20º as the accuracy of ICESat-2 is reduced in such cases (Brunt et al., 2019; Enderlin et al., 2022; Hao et al., 2022). Finally, penetration bias values were filtered using the interquartile range, defined by the first and third quartiles, to establish lower and upper bounds (Equation (1) and (2)).

$$\text{Lower Bound} = Q_1 - 1.5 \times \text{IQR} \qquad (1)$$


$$\text{Upper Bound} = Q_3 + 1.5 \times \text{IQR} \qquad (2)$$

Where, $Q_1$ is first quartile of the penetration bias values, $Q_3$ is third quartile of the penetration bias value and IQR = Interquartile range.

This approach effectively removes outliers in the penetration bias distribution, resulting in a dataset comprising 276,549 points (Figure 4). Atmospheric features from ECMWF ERA-5 datasets (Table 1) are added in the CSV file for each

data point. The dataset available on the ECMWF website is NetCDF file on an hourly basis for each day from the year 1944 onwards and is updated daily. We use the ECMWF website API to download NetCDF files for each date and time of the





TanDEM-X DEM height. For the atmospheric feature of wind speed, the horizontal (u) and vertical (v) components are obtained and subsequently used to calculate the wind speed magnitude and direction.

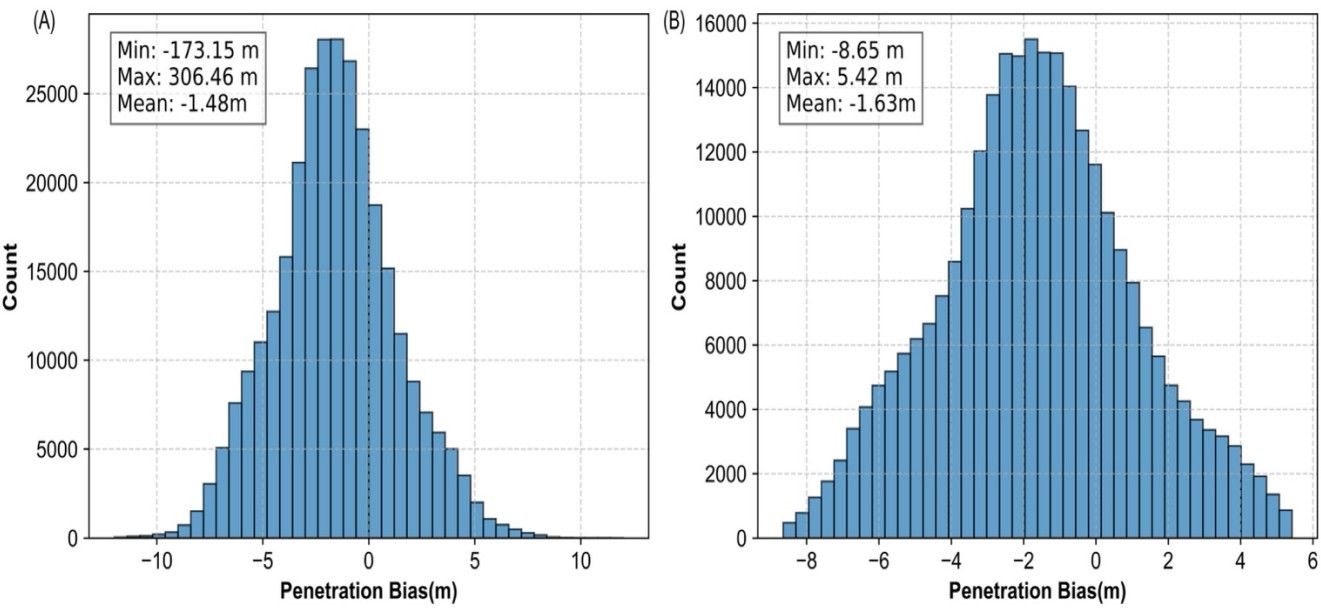

**Figure 4. Penetration Bias values before (left) and after (right) dataset preparation. By applying the filter, the extreme positive and negative outliers in the penetration bias are removed. 'Min' stands for Minimum, 'Max' stands for Maximum values of differences between TDX and ICESat-2. Negative values mean TDX DEM height below ICESat-2. Positive values can be observed due to noise of the InSAR measurement.**

### 3.2 Machine Learning Algorithms and data preparation

Before data feeding in the machine learning and deep learning algorithms, data is divided into training, validation and testing data. So, data was split, into training (64%), validation (16%), and test (20%) sets using an initial split of 80%-20% for training and test data, followed by a secondary split of the training set for validation data. Random shuffling is applied while data splitting to ensure that data from all glaciers gets distributed across each subset, thereby enabling the model to learn generalized patterns across all glaciers. This approach prevents overfitting to any specific glacier and ensures model generalization to learn broad patterns in the data. Further, standardization for each variable is achieved using a standard scaler (Swarna Priya et al., 2020;Sharma, 2022).

We compare 14 algorithms implementing non-linear regression to identify patterns between radar penetration bias, SAR and atmospheric features (Table1). These algorithms include five machine learning algorithms, namely Random Forest (RF), light Gradient Boosting Machine (GBM), Extreme Gradient Boosting (XGBoost), Decision Tree, K-Nearest Neighbours (KNN) and nine deep learning algorithms including, Fully connected NN, Batch Normalization NN, Leaky ReLU NN, Residual Network (ResNet), Deep Neural Network (DNN), Transformer Multi-Head Attention NN, Auto Encoder NN, Multi-



layer Perceptron (MLP) NN and Wide & Deep NN. The accuracy of all these machine learning and neural network models are compared and further analysis for the best feature assessment is also carried out. Details of the model architectures used in this work are listed in Table 3. The Learning rates are adjusted iteratively, and an appropriate learning rate values are chosen to optimize performances of each model.

**Table 3. Details of parameters in machine learning and Deep learning algorithms defined for model architecture**

| Method | Model architecture parameters in this work |
|---|---|
| Random Forest (Breiman, 2001) | Number of trees = 150, Maximum depth of trees=20, python module is scikit learn |
| Decision Tree (Quinlan, 1986) | Maximum depth of the tree = 15, Minimum number of samples required to split an internal node=10, Minimum number of samples required to be at a leaf node=5, criterion function to measure the quality of a split= 'squared_error', python module is scikit learn |
| KNN (Cover & Hart, 1952) | Number of neighbors = 7, Metric used for distance computation= 'manhattan', Weight function used in prediction= 'distance', python module is scikit learn |
| Light GBM (Ke et al., 2017; Shehadeh et al., 2021) | Objective task= 'regression', Metric to be evaluated on the evaluation set= 'mse', Type of boosting algorithm= 'gbdt'(Gradient Boosting Decision Tree), Maximum number of leaves in one tree= 64, To select percentage of features before training each tree=0.9, L2 regularization= 0.1, Number of iteration=500, python module lightgbm |
| XG Boost (T. Chen & Guestrin, 2016; J. Dong et al., 2022) | Objective task: 'reg:squarederror'(Regression with squared loss), Evaluation metrics for validation data= 'rmse', Maximum depth of a tree=6, Subsample ratio of columns when constructing each tree=0.8, Subsample ratio of the training instances=0.8, Number of boosting iterations=500, python module xgboost. |
| Fully Connected NN (Rumelhart et al., 1986; X. Dong et al., 2024) | 2 hidden layers with neurons 64 and 32, Activation function=tanh, epoch=100, Number of training samples=64, Optimizer= Adam, Tensor flow architecture in python |
| Batch Normalization NN (Ioffe & Szegedy, 2015) | 2 hidden layers with neurons 128 (connected with input layer) and 64, Activation function=ReLU, epoch=100, Number of training samples=64, Optimizer= Adam, Tensor flow architecture in python |



| Leaky ReLU NN (Maas et al., 2013; Maniatopoulos & Mitianoudis, 2021) | 2 hidden layers with neurons 128(connected with input layer) and 64, Activation function=ReLU, Epoch=100, Number of training samples=64, Optimizer= Adam, Tensor flow architecture in python |
|---|---|
| ResNet (He et al., 2015; Hanif & Bilal, 2020) | 4 hidden layers with neurons128 (connected with input layer), 128, 64 and 32, 1 residual connection with 128 neurons, Activation function=ReLU, Epoch=100, Number of training samples=64, Optimizer= Adam, Tensor flow architecture in python |
| DNN (Rumelhart David E. et al., 1986; Montavon et al., 2018) | 6 hidden layers with neurons 1024(connected with input layer), 512,256,128,64,32, Activation function=ReLU, Epoch=100, Number of training samples=256, Batch normalization with drop out of 0.2 in the first two layers and 0.1 the next two layers, Regularization applied to the layer's weight matrix(kernel)=l2(0.001), Optimizer= Adam, Tensor flow architecture in python |
| Transformer Attention NN (Vaswani et al., 2017) | 2 hidden layers in feed forward network with neurons 128, 64, 3 hidden layer with neurons 64(connected with input layer),64,32 epoch=100, Number of training samples=64, Activation function=ReLU, Optimizer=Adam, Tensor flow architecture in python |
| Auto Encoder NN (Hilton G.E & Salakhutdinov R.R, 2006) | Encoder with neurons, 64 for the input and 32 in the hidden layer and 64 for the decoder, Activation function=ReLU, Epoch=100, Number of training samples=64, Optimizer= Adam, Tensor flow architecture in python |
| MLP (Minsky Marvin L. & Papert Seymour A., 1969) | 4 hidden layers with neurons 256(connected with input layer) 128, 64 and 32, drop out of 0.2 with batch normalization, Activation function=ReLU, Epoch=50, Number of training samples=64, Optimizer= Adam, Tensor flow architecture in python |
| Wide and Deep NN (Cheng et al., 2016; Radhakrishnan et al., 2023) | 1 layer with 32 neurons for wide NN, 4 hidden layers in Deep NN with neurons 256(connected with input layer), 128,64 and 32, Activation function=ReLU, Epoch=100, Number of training samples=128, Optimizer= Adam, Tensor flow architecture in python |




### 3.3 Input and running of the data models

To understand potential interrelations between radar and atmospheric features we focus to plot histograms of our variables and
calculate their respective correlation (Figure 5, Figure 6). Obviously, elevation values of both TanDEM-X and ICESat-2 are
showing high correlation. Various environmental features like Ultraviolet Albedo and Near Infrared Albedo along with snow
depth show high values of correlation. Whereas features like snow density shows negative correlation with snow albedo values.
Figure 6 provides a comprehensive overview of the interrelations among the features which are showing high variations in
Figure 5 and selected for the model input. Based on this analysis we decide to remove some of the highly correlated variables
and leave only seven features including temperature of the air at 2m above the land surface, snowfall, surface pressure, snow
density, snow evaporation, wind speed and wind direction. Removing the features that do not show much variation in the
histograms (Figure 5), helps reduce redundancy, improve learning efficiency, and mitigate risks like overfitting and
optimization challenges (Halkjrer & Winther, 1996). However, for the elevation inputs at the data point, TanDEM-X DEM are
used in the data models, while ICESat-2 elevation data is excluded.








**Figure 5. Histograms of the final filtered data for all radar and atmospheric features for the ICESat-2 data points. Features namely NIR albedo, Snow depth, Temperature of snow layer, total column snow water, total precipitation, snow albedo, UV visible albedo are removed from the input features to the models because either it is not varying much or because it is lightly correlated with other features. ICESat-2 elevation data is not used as input; only TanDEM-X DEM values are considered for the AI models.**






**Figure 6. Correlation map of the radar and atmospheric features showing high variation in Figure 5. ICESat-2 elevation**

**data is not used as input; only TanDEM-X DEM values are considered for the AI models.**





## 4 Results

We run our machine learning architectures on four total test cases as given in table 4. First, we train our network only on SAR features and then we train the same architectures on all the SAR and ECMWF features combined. Each of these tests is performed using only TanDEM-X data acquired within 15 and 30 days of the corresponding ICESat-2 acquisition. These
temporal windows are selected to evaluate the algorithm's sensitivity under varying time intervals, simulating potential operational scenarios. Given that the data were collected during the winter season, when passive optical sensors are generally ineffective due to low solar illumination in Antarctica, minimal changes in glacier surface are expected during this period.

**Table 4 Details of test cases used for training data for models.**

| Test Case | Training Data |
|---|---|
| Test Case1 | Radar Features with 30 days' time difference between TanDEM-X and ICESat-2 elevation data |
| Test Case 2 | Radar Features with 15 days' time difference between TanDEM-X and ICESat-2 elevation data |
| Test Case 3 | Radar and Environmental Features with 30 days' time difference data between TanDEM-X and ICESat-2 elevation data |
| Test Case 4 | Radar and Environmental Features with 15 days' time difference between TanDEM-X and ICESat-2 elevation data |

### 4.1 Only SAR features as input to the data models

Initially, all the models are run by considering only 6 radar features namely TDX DEM height, coherence, amplitude, height of ambiguity, local incidence angle and slope as taken as input with penetration bias as the target variable. For 30 days' time difference, number of data points as input are 276,549 which gets reduced to 215,620 when considering 15 days' time interval. Radar features are considered as sole input in the data models to analyse their impact on penetration bias. The result parameters
of all the models are given in Table 5 for 30 days (Test Case 1 in Table 4) and 15 days runs (Test Case 2 in Table 4) respectively. Model predictor parameter values presented in Tables 5 indicate that the differences in RMSE, standard deviation, mean, and median error between the 30-day and 15-day datasets are, on average, within 2 cm across all algorithms. Notably, the minimum and maximum errors are, on average, approximately 40 cm smaller for the 15-day temporal difference. A visual analysis of the same results is shown in Figure S1 of the Supplement. Here we observe that, except for Auto Encoder NN, all the models
show slightly better values for the 15 days' time interval data. Among these models, Random Forest is the best performed model which shows the lowest RMSE close to 1.2 m and MAE of 0.8 m along with highest values of R2 score of 0.8 for both the test data types.

    The Auto Encoder is the least effective model for both the types of test datasets. The reason is that Auto Encoders are mainly applied for unsupervised learning tasks like dimensionality reduction and anomaly detection by reconstructing the
input data and not inherently optimized for regression tasks. In this work, Auto Encoder network was first trained to compress



and reconstruct the input features and then the encoded representations are used for regression task. So, it may be that this indirect approach for predicting penetration biases is resulting into relatively higher errors than the rest of the algorithms. Similarly, Decision Tree is the least effective model among machine learning models which is giving high RMSE value nearer to 1.5 and MAE value as 1.08 for both 30 days and 15 days' time interval (Figure S1 of the Supplement). The reason may be

cited as Decision Tree regression model consists of single tree which makes decision based on threshold values of features. However, it's not much effective when there is a complex interaction between the features of the input data. Whereas Random Forest which is an ensemble of multiple decision trees is very effective in complex non-linear regression tasks. This is evident in the comparison of model prediction parameters (Table 5 and Figure S1 of the Supplement).

**Table 5 Model prediction parameters of DNN and Random Forest models all the test cases in Table 4. All errors are in**
**meters except for R2 score and explained variance.**

| Name of the Model | RMSE | MAE | R2 Score | Explained Variance | Error Min | Error Max | Mean Error | Median Error | Std Dev of Error |
|---|---|---|---|---|---|---|---|---|---|
| For Radar feature as input and time difference = 30 days | | | | | | | | | |
| Random Forest | 1.18 | 0.83 | 0.81 | 0.81 | -8.79 | 9.08 | 0.01 | -0.02 | 1.18 |
| DNN | 1.26 | 0.90 | 0.78 | 0.78 | -8.64 | 9.83 | -0.01 | -0.02 | 1.26 |
| For Radar feature as input and time difference = 15 days | | | | | | | | | |
| Random Forest | 1.18 | 0.82 | 0.82 | 0.82 | -8.56 | 9.14 | 0.01 | -0.01 | 1.18 |
| DNN | 1.21 | 0.86 | 0.81 | 0.81 | -8.67 | 9.42 | 0.03 | 0.02 | 1.21 |
| For both Radar and Atmospheric features as input and time difference = 30 days | | | | | | | | | |
| DNN | 1.04 | 0.74 | 0.85 | 0.85 | -8.75 | 8.95 | 0.03 | 0.02 | 1.04 |
| Random Forest | 1.08 | 0.77 | 0.84 | 0.84 | -8.60 | 8.79 | 0.03 | 0.02 | 1.08 |
| For both Radar and Atmospheric features as input and time difference = 15 days | | | | | | | | | |
| DNN | 1.03 | 0.74 | 0.86 | 0.86 | -8.23 | 9.08 | -0.01 | 0.00 | 1.03 |
| Random Forest | 1.09 | 0.78 | 0.84 | 0.84 | -9.26 | 9.32 | 0.03 | 0.02 | 1.09 |

### 4.2 Using both SAR and ECMWF features for training

Along with the SAR features, we add seven atmospheric features including, temperature of the air at 2 m above the land surface, snowfall, surface pressure, snow density, snow evaporation, wind speed and wind direction (Table 1).

395       Table 5 the model prediction parameters for the input data with a time difference of 30 days or Test Case 3 (Table 4) shows that DNN is the best performing models among all the data models with RMSE almost equal to 1.04 m, MAE equal to 0.74 m along with high corresponding values of R2 score and explained variance. Figure 7 shows the density scatterplot with an $R^2$ score of 0.85 and histogram of differences of the DNN model for 30 days' time interval data. The histogram of errors illustrates a gaussian distribution of errors suggesting that errors are symmetrically distributed, and the model is able to capture





the non-linear features in the dataset and correct for any systematic bias. Overall, after incorporating additional ECMWF features all error metrics are reduced for all models and RMSE is reduced between 10 to 20 cm. Similarly, Table 5 the model prediction parameters for the input data with a time difference of 15 days or Test Case 4 (Table 4) also shows that DNN is the best performing model with RMSE equal to 1.03, MAE equal to 0.74 with R2 score and explained variance as 0.86. Interestingly the accuracy between the 30- and 15-days' time acquisition interval between TDX and ICESat-2 measurements

does not influence the accuracy of the results when including ECMWF variables. Figure 9 shows the model analysis of DNN for 15 days' time interval data, there is a minor increase in performance as compared to the model parameters for 30 days' time interval data (Figure 7) are used as input. Overall, RMSE and MAE have been reduced between 10 and 20 cm and R2 score along with explained variance values improved after including ECMWF atmospheric features as inputs to the NN algorithms.

410        Figure S2 of the Supplement shows the comparison of the errors for 30days' time interval and 15 days' time interval input data. It may be observed that all the errors are reduced for most of the models. However, decision tree and Auto Encoder NN are least effective models for both datasets. Additionally, Random Forest has also shown good results and emerged as a robust algorithm for non-linear regression with RMSE as 1.08m, MAE as 0.77m and high R2 score along with explained variance as 0.84 for 30 days and nearly the same model prediction parameters for 15 days' time difference data (Table 5).

Figure 8 and Figure 10 show the model analysis of Random Forest model for 30 days and 15 days' time interval data. Figure S3 and Figure S4 of the Supplement, show a comparison of model parameters when only radar features and both radar and atmospheric features are used as input for 30 days and 15 days' time interval data. Both Figure S3 and Figure S4 of the Supplement, shows how the model accuracies improve when both radar and atmospheric features are used as input to the models.

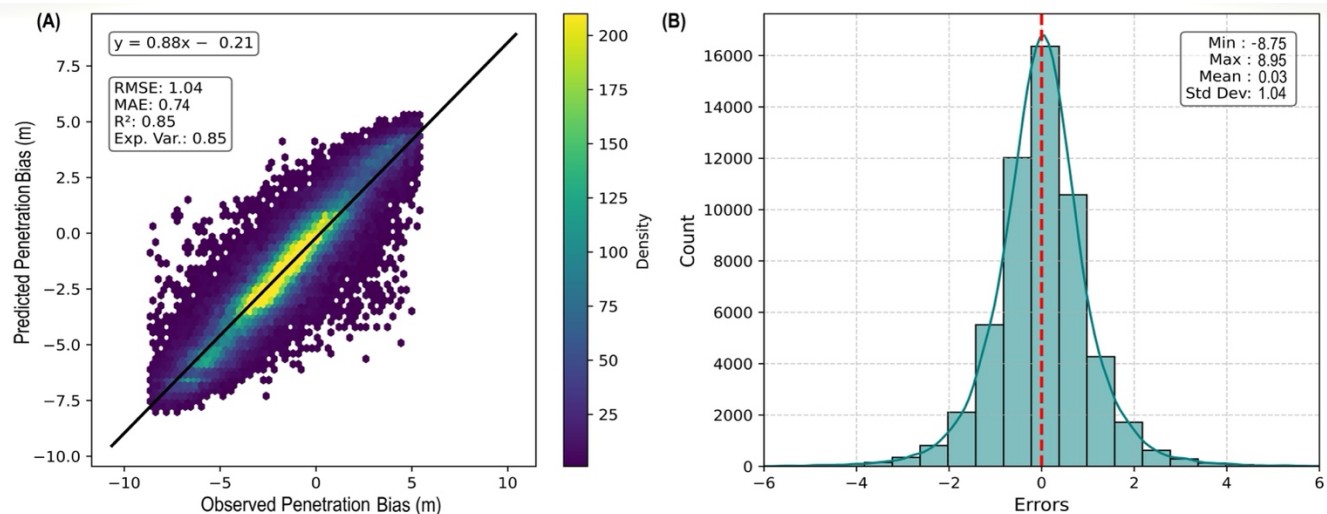


**Figure 7. DNN model analysis for both Radar and Atmospheric features as input data for time difference=30days (A) Error parameters in model prediction (B) Histogram of Errors.**



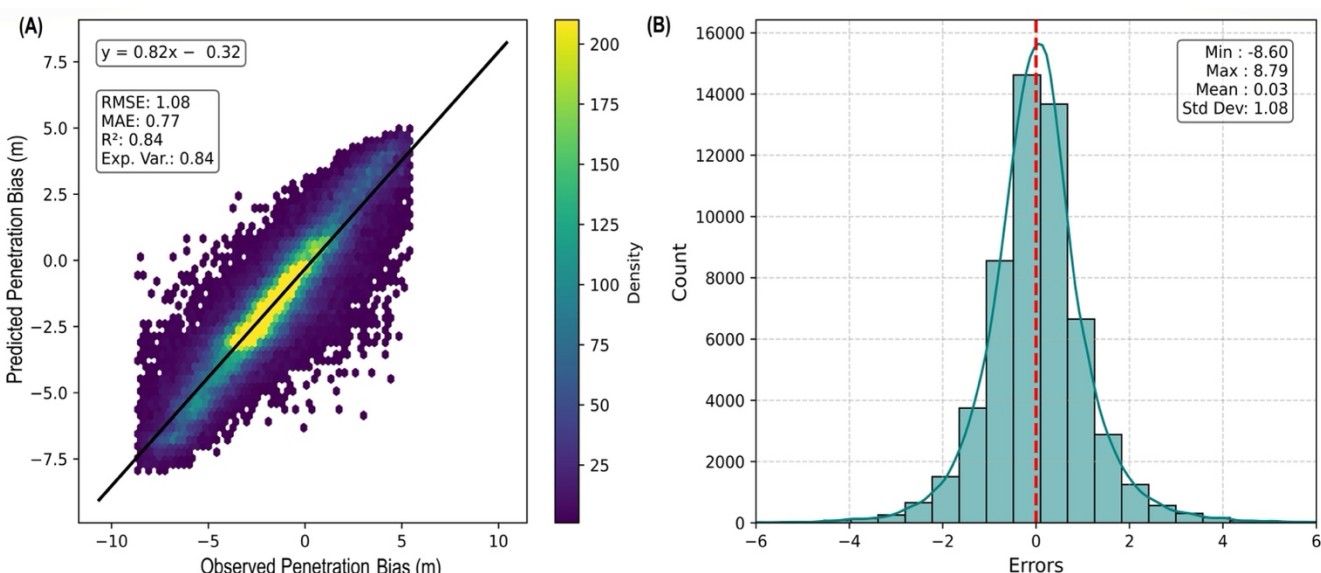

**Figure 8. Random Forest model analysis for both Radar and Atmospheric features as input data for time difference=30 days (A) Error parameters in model prediction (B) Histogram of Errors.**

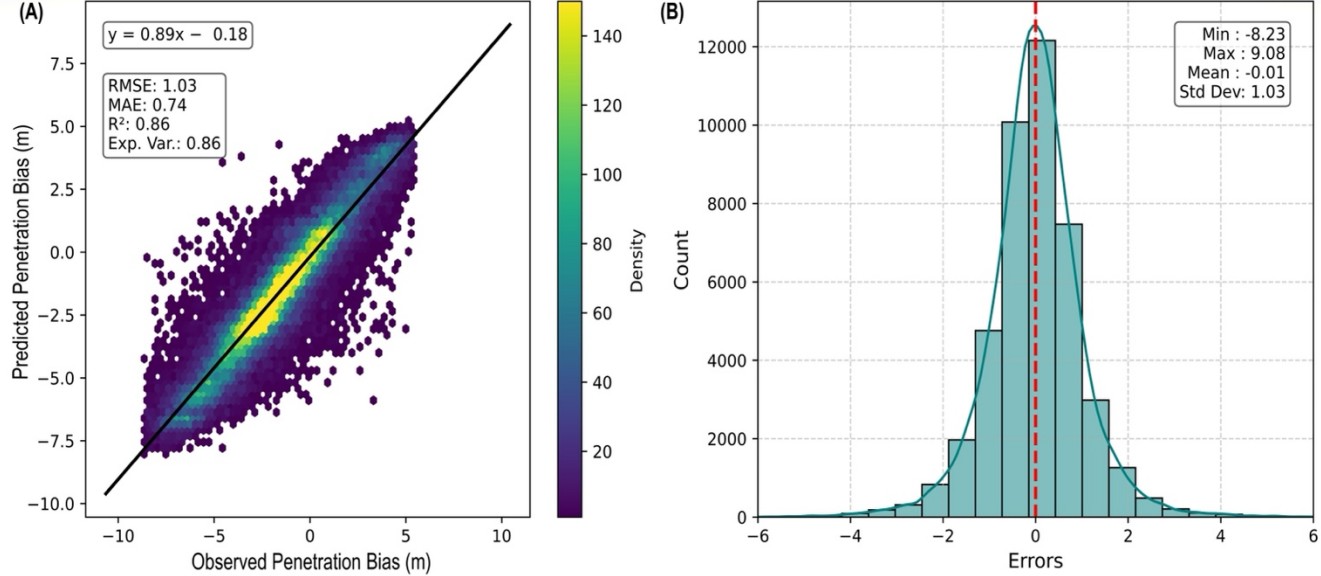

**Figure 9. DNN model analysis for both Radar and Atmospheric features as input data for time difference=15days (A) Error parameters in model prediction (B) Histogram of Errors.**





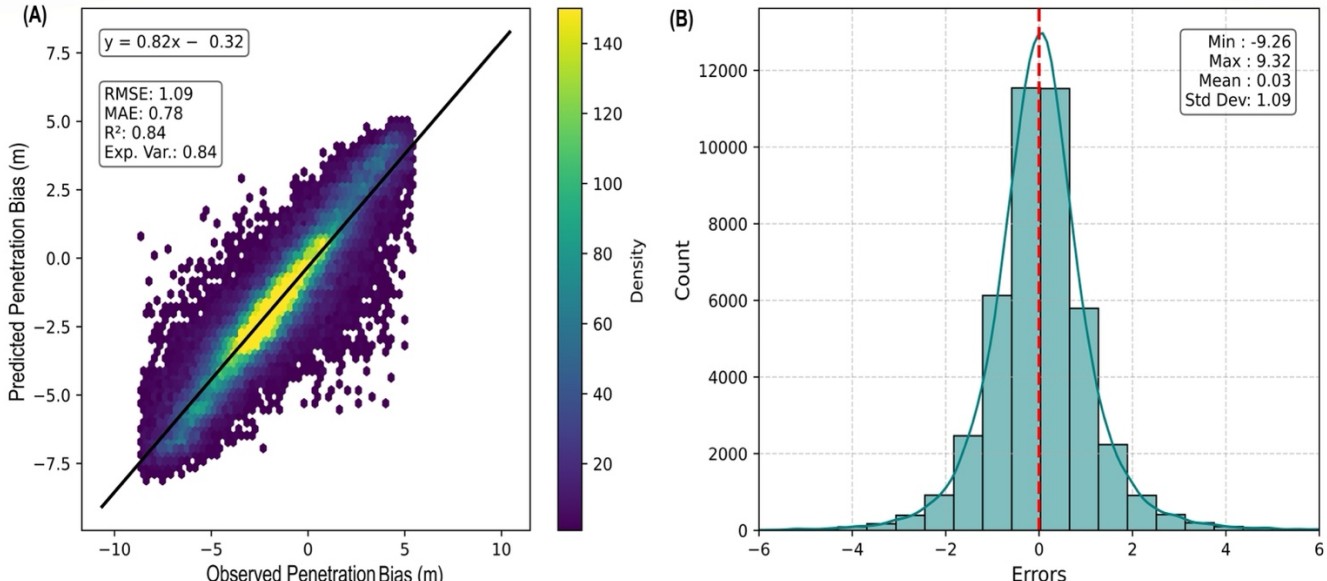


**Figure 10. Random Forest model analysis for both Radar and Atmospheric features as input data for time difference=15days (A) Error parameters in model prediction (B) Histogram of Errors.**

### 4.2.1 Shapley Additive exPlanations (SHAP) Analysis of the DNN model

As the DNN model with model architecture given in Figure 11 is most efficient when both radar and atmospheric features are taken as input in the data models, a SHAP analysis (Lundberg & Lee, 2017) is conducted to identify the dominant features which are influencing the model's prediction. A SHAP analysis plot helps interpret how individual features influence the predictions of a machine learning model. In a typical SHAP summary plot (Figure 12 and Figure 13), the horizontal position of each dot indicates the feature's impact on the prediction, where values to the right (greater than zero) show a positive

contribution, increasing the model's predicted outcome, while values to the left (less than zero) indicate a negative contribution, reducing the predicted outcome. Features are arranged vertically, ordered from the most influential at the top to the least influential at the bottom. Each dot represents a single data point, with colours illustrating feature values: red indicates high values and blue indicates low values. Clusters or density of dots reveal common patterns or typical behaviours in how features influence predictions. For example, a feature with predominantly red dots to the right suggests that higher values of that feature

consistently increase predictions.

Additionally, SHAP dependence plots provide deeper insights by plotting feature values on the x-axis against their SHAP values on the y-axis, revealing potential nonlinear relationships or interactions. Interaction effects with other features are visualized through color coding, indicating how the relationship between a feature and its influence changes under varying conditions of another feature. For instance, a rising trend of SHAP values as the feature value increases indicates a positive




relationship. This analysis is especially valuable for our dataset including both radar and non-radar features as it aids in refining the training dataset.

In order to reduce the computational load, the SHAP analysis is carried out for 1000 random samples from each feature in the dataset and summary plots are generated as given in the Figure 12 for 15 days and Figure 13 for 30 days' time interval data. The top five features which are influencing the DNN model prediction ability includes four radar features, in
order, coherence, amplitude, height of ambiguity, TanDEM-X DEM height values and one ECMWF feature as 2 m temperature. The least effecting feature is snow density. Other significant atmospheric features which are affecting the DNN model includes wind speed, wind direction and surface pressure which also shows broad data distribution in figure 5. For 30 days data the top five features effecting the model are shown in order (Figure 13). The top performing features is coherence followed by amplitude. The least affecting factor remains snow density for both the types of datasets. We also conducted a
SHAP analysis for the Random Forest which is the second-best performing regression model after DNN. The SHAP analysis of the random forest gave the same the result of the DNN model for both 15 and 30 days.

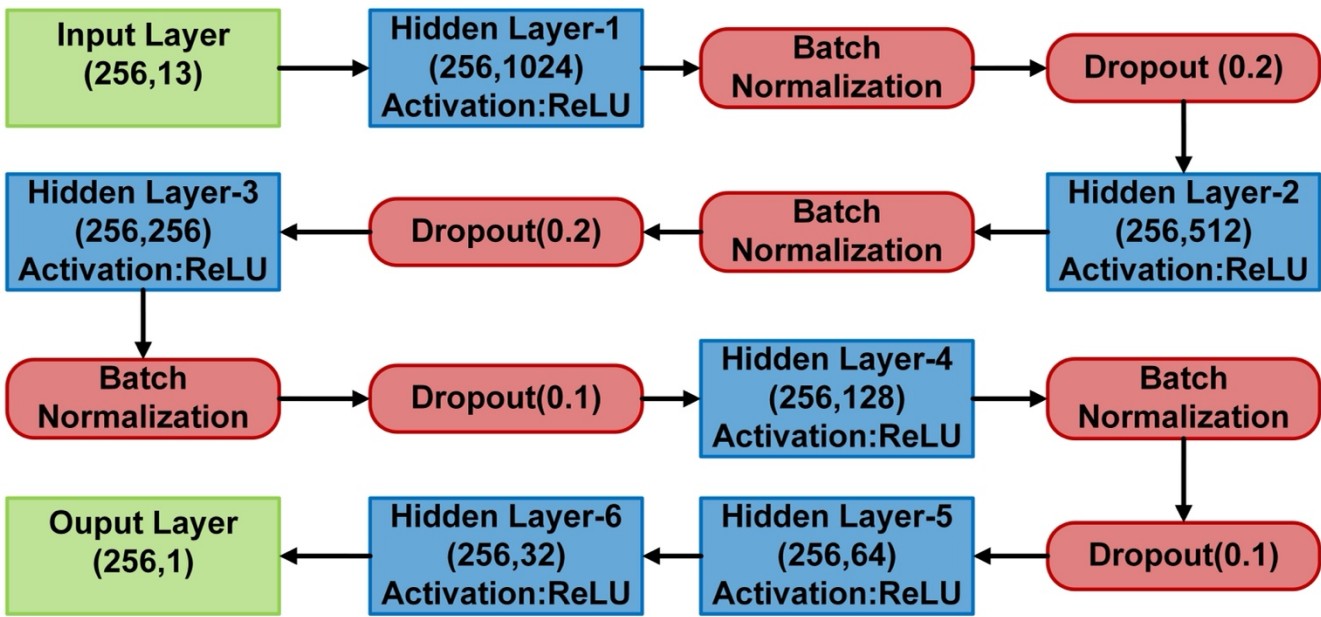

**Figure 11. DNN architecture block diagram, with batch size as 256, optimizer as Adam and l2 kernel regularization at**
**each hidden layer.**





**Figure 12. SHAP Analysis for 15 days' time interval data (A) Summary plot (B) Feature importance plot for DNN model.**






**Figure 13. SHAP Analysis for 30 days' time interval data (A) Summary plot (B) Feature importance plot for DNN model.**



## 5 Discussion
### 5.1 Elevation Accuracy of 1 m: Glaciological Significance

The bias-corrected TanDEM-X DEM height achieve an RMSE on the order of 1 m when compared to ICESat-2, marking a substantial improvement over uncorrected X-band DEMs. Previous assessments showed that raw TanDEM-X heights in Antarctica were systematically low by a few meters relative to laser altimetry with a mean of about ~3.2 m lower due to radar penetration (Wessel et al., 2021). In the cold, dry interior of Antarctica, penetration biases can reach 5 m or more, pushing 90% vertical errors to ~6.3 m (LE90) (Wessel et al., 2021). Achieving ~1 m RMSE and a mean of 1 cm after correction thus

effectively eliminates the bulk of this bias, bringing TanDEM-X accuracy in line with laser altimetry and stereophotogrammetric DEMs. Notably, this precision approaches the in-situ validation limits; for example, even within a firn zone used for block-wise calibration, residual elevation scatter was ~0.9–1.2 m (Wessel et al., 2021).

In glaciological terms, a 1 m elevation uncertainty is comparatively small: it enables detection of subtle elevation changes that are on the order of annual accumulation or sub-decadal ice thinning. This improved accuracy strengthens

confidence in using X-band InSAR data for ice-sheet studies, as the corrected DEM can be treated nearly as "surface-equivalent," free from the multi-meter biases that previously complicated interpretations (Wessel et al., 2021). For example, surface elevation changes, flow dynamics, and grounding line positions derived from the DEM will be more reliable, and mosaic DEM products (like the TanDEM-X PolarDEM) can be integrated with other datasets without large bias corrections (Kim & Kim, 2017; Milillo et al., 2019; Milillo et al., 2022) Overall, reaching 1 m RMSE represents a significant step toward

high-precision elevation models in Antarctica, enabling more accurate ice-sheet topography mapping and change detection needed for climate and sea-level assessments (Wessel et al., 2021).

### 5.2 Impact on Ice Mass Balance and Melt Rate Estimates

Improving the X-band DEM accuracy has direct implications for melt rates and ice mass balance calculations. Elevation change

is a primary observable for geodetic mass balance and melt rates. Biases of several meters, if uncorrected, can severely skew volume change estimates(Milillo et al 2019; Brancato et al., 2020; Wessel et al., 2021; Milillo et al., 2022). By correcting the penetration bias, the TanDEM-X data can be used to measure true surface elevation changes rather than artifacts of varying snow conditions. This is critical because even modest systematic errors can translate into large mass errors when integrated over vast areas. For instance, a 1 m average elevation bias across an entire ice sheet sector would misrepresent ice volume by

billions of cubic meters. With the bias correction, the TDX DEM differencing between epochs yields more accurate ice thinning or thickening magnitudes. Consequently, estimates of mass loss or gain are more accurate, refining assessments of Antarctica's contribution to sea-level rise. Furthermore, the ability to attain ~1 m elevation accuracy without the presence of data gaps as for the ICESat-2 acquisition scheme means that shorter-term or smaller-scale changes can be resolved with confidence.

This is especially important for regions experiencing rapid dynamics: for example, outlet glaciers, grounding lines and ice shelves often exhibit thinning on the order of a few meters per year (or more in extreme cases) (Milillo et al., 2019;



Brancato et al., 2020; Milillo et al., 2022; Ciracì et al., 2023). A bias of several meters could mask or falsely amplify such changes, but with corrected data, observed elevation changes can be directly attributed to physical processes (e.g. ice dynamics or surface melt) rather than sensor bias. Notably, in radar percolation zones where seasonal melt occurs, uncorrected InSAR
measurements can differ by up to 10–15 m from the actual surface (Wessel et al., 2021) .This is an error large enough to invert the sign of a mass balance estimate. Our corrections eliminate these ambiguities: for example, a near-zero net bias was achieved (mean bias ~0.01 m), so any measured lowering (or rise) in surface elevation reflects genuine mass loss (or gain) rather than radar penetration variability.

In summary, bias-corrected bistatic InSAR data provide a solid foundation for calculating volume changes and mass
balance. They ensure that glaciological interpretations, such as estimating mass loss from elevation time series or calibrating firn compaction models, are based on real surface changes which reduces one of the long-standing uncertainties in radar remote sensing of ice sheets (Wessel et al., 2021) and enhances our ability to close regional mass budgets with confidence.

**5.3 SHAP Feature Importance: Coherence vs Amplitude and Dataset Differences**

To interpret the trained neural network, we applied SHAP analysis, which revealed that interferometric coherence and SAR backscatter amplitude are among the top predictors of penetration bias. Interestingly, we observed a slight shift in their relative importance depending on the time window of data used. For the dataset where ICESat-2 and TanDEM-X acquisitions were within 30 days (Figure 13), coherence was the most influential feature, whereas for a subset with shorter 15-day intervals (Figure 12), the SAR amplitude ranked marginally higher. Physically, both coherence and amplitude carry information about
the snowpack and scattering processes, and our results suggest they are almost equally critical. InSAR Coherence reflects volume scattering and temporal stability: a lower coherence generally indicates more signal penetration into the snow volume or surface changes over time, both of which correspond to a larger elevation bias (Bamler & Hartl, 1998; Wessel et al., 2021). SAR Amplitude (or backscatter intensity), on the other hand, indicates surface reflectivity: a stronger X-band return often implies a denser or rougher surface that backscatters more energy near the surface (hence less penetration), whereas weak
returns can imply softer, low-density snow or smoother surfaces that allow deeper penetration (Wessel et al., 2021).

The SHAP analysis suggests that over a 30-day interval, coherence slightly outperforms amplitude in predicting bias. This likely owes to the amplitude sensitivity to any surface changes that might occur over a longer period between LIDAR and InSAR observations. In bistatic acquisitions, coherence can be reduced by volume scattering from deep layers; over 30 days, additional events could occur from events like snowfall or wind redistribution, making coherence a strong indicator of
those penetration-altering events.

In contrast, for closely timed acquisitions within 15 days, the snow surface is less likely to have changed. In this scenario, amplitude differences become a clearer discriminator of penetration bias (essentially serving as a proxy for different snow facies or densities) (Wessel et al., 2021), hence its slightly higher importance. However, it is important to emphasize that the differences in SHAP values between coherence and amplitude were very small. Given the limited sample size (only ~1,000
data points used in the SHAP evaluation) and the inherently noisy nature of feature attribution in complex models, we should



be cautious not to over-interpret which feature "wins" in each case. The near parity of coherence and amplitude in our analysis indicates that both features complement each other in predicting the bias.

This is a finding consistent with previous studies that used combinations of interferometric correlation and backscatter to empirically model radar penetration (Fischer et al., 2019; Fischer et al., 2020;Wessel et al., 2021;Dall et al., 2001). For

example, a study in northern Greenland found that incorporating both coherence and intensity in a regression improved bias estimates (Abdullahi et al., 2019)underscoring that encapsulates a different aspect of the scattering physics. Our SHAP results broadly corroborate this: coherence excels when capturing volume scattering, while amplitude captures persistent surface characteristics, they are both essential. The slight discrepancy between the 15-day and 30-day feature rankings could simply reflect the sample variance or specific conditions of those subsets. With a larger dataset and more diverse conditions, one might

expect coherence and amplitude to share nearly equal importance, as both are fundamentally linked to the radar elevation bias. In summary, the SHAP analysis confirms the dominant role of SAR signal metrics (coherence and amplitude) in bias prediction, followed by the atmospheric factors discussed above. The differences observed between the two time-interval subsets likely do not indicate a fundamentally different physical regime, but rather the context in which each feature has the most opportunity to express variability (temporal vs. spatial/structural differences in the snowpack). Our interpretation is that

coherence and amplitude are jointly the best predictors, and minor swaps in their ranking reflect marginal conditions or noise. Therefore, any operational model should retain both features.

The results also demonstrate the value of explainable AI techniques in geoscience: they help ensure our model's behaviour aligns with known physics (it makes sense that coherence and amplitude are top features, as observed), and they highlight where more data might be needed to resolve feature importance with greater confidence.


**5.4 Influence of Atmospheric and Snow Variables on Bias Prediction**

Incorporating ECMWF atmospheric reanalysis variables was found to improve the penetration bias predictions by an additional ~10–20 cm, a meaningful gain in accuracy. The most influential predictors identified were 2 m air temperature, near-surface wind speed and direction, and surface pressure, whereas variables like modelled snow density were notably less useful.

The prominence of 2 m temperature is consistent with physical expectations: colder air temperatures generally indicate cold snow/firn conditions that permit deeper X-band penetration, whereas near-freezing or melt conditions lead to shallow penetration due to increased absorption and possible ice crust formation (Wessel et al., 2021). The model likely captures this by using air temperature as a proxy for snowpack wetness or the presence of refrozen layers.

Wind speed and direction emerged also as the second most important feature because wind redistributes snow and

alters surface properties. Strong winds can sublimate and compact the surface, creating wind crusts or glazed areas of high density; they can also form sastrugi and other anisotropic roughness features aligned with the wind. These processes affect the radar signal: a dense, wind-packed surface layer produces a high backscatter and shallow penetration (acting as a bright reflective layer) (Wessel et al., 2021), while freshly deposited, low-density snow (often in wind-sheltered areas) leads to deeper penetration. The importance of wind direction suggests that the relative orientation of surface features relative to the radar look



angle can modulate the backscatter and coherence. Indeed, extensive "wind glaze" areas on the East Antarctic plateau, formed on leeward slopes under persistent katabatic winds, are known to have unique scattering properties (large grains, smooth surfaces) that influence radar returns (Scambos et al., 2012).

Surface pressure's contribution is likely indirect; atmospheric pressure patterns are linked to weather events (e.g. low-pressure systems bring snowfall and warmer air, high-pressure brings clear, cold conditions). Thus, surface pressure can serve
as a broad indicator of recent storm activity or calm conditions, complementing the temperature and wind information in characterizing the snow state.

In contrast, the reanalysis snow density variable was among the least informative features. This may be because first the ECMW values are all very close to 300 kg/m$^3$ (Figure 5) and second because the modelled snow/firn density is too smooth or climatological to reflect the specific stratigraphy and surface conditions that determine radar penetration. In polar deserts
like East Antarctica, the near-surface density tends to evolve slowly and may not capture ephemeral phenomena (such as a thin crust or a layer of hoar) that strongly affect X-band scattering.

Moreover, the spatial resolution of ECMWF (~10–30 km for modern reanalysis) means local variability is lost, whereas pointwise predictors like wind and temperature anomalies can still proxy sub-grid effects (e.g. strong katabatic winds over a particular slope). Our findings align with the notion that dynamic atmospheric conditions leave an imprint on the
snowpack's electromagnetic properties. Variables that describe the current state of the surface (temperature affecting dielectric losses, wind shaping surface texture) offer more predictive power than bulk snow properties from the model. In essence, a cold, calm period versus a warm, windy storm will produce very different penetration biases, and our model learns these differences. The minor (~0.1–0.2 m) yet significant boost in accuracy from including these variables highlights the value of environmental data in bias correction. It demonstrates a multi-disciplinary synergy: by fusing meteorological information with
remote sensing, we capture subtle effects such as thin crusts or wind slabs that a purely radar-based approach might miss. This result also suggests that future penetration models could benefit from even more detailed snowpack characterization (e.g. from regional climate models or field measurements), although our results indicate diminishing returns beyond the key features of temperature and wind. Finally, the low importance of snow density underscores the need for improved snow/firn modeling in polar regions, current models may not represent the features most relevant to X-band radar, such as layering or grain size,
reinforcing the value of using observable proxies like temperature and wind in the meantime.

**5.5 Implications for NASA's STV Mission: SAR Interferometry vs. Lidar Altimetry**

These findings carry significant implications for the design of future ice-monitoring missions, particularly the NASA's Surface Topography and Vegetation (STV) study (Andrea & Craig, 2024). A core challenge for STV in the cryosphere will be obtaining
high-resolution, time-evolving maps of ice surface elevation over vast areas, including fast-changing zones such as ice sheet margins and glacier termini.



Our study demonstrates that SAR bistatic interferometry is a viable technique to meet this challenge, provided that radar penetration biases are properly accounted for. The ability to retrieve surface elevations with ~1 m accuracy from TanDEM-X bistatic data confirms that SAR can serve as a powerful tool for measuring ice topography, rivalling the accuracy of laser altimeters while offering superior spatial coverage. Crucially, SAR interferometry produces spatially continuous elevation maps. This is especially advantageous in regions like ice sheet grounding zones, or Greenland outlets where capturing continuous elevation changes is essential. Grounding zones often span several kilometres of complex terrain where ice goes afloat, involving sharp elevation gradients and localized deformation (Ciracì et al., 2023).

A swath-based SAR measurement can map the entire grounding zone in one pass, revealing the full pattern of elevation change (for example, showing a continuous flexure bulge or the line of tidal uplift). In contrast, lidar altimetry provides point or profile measurements along discrete ground tracks. Even a dense altimetry mission like ICESat-2, with multiple beams, has inter-track gaps on the order of a few kilometres at high latitudes. Important features can fall between tracks, and the spatial continuity must then be reconstructed via interpolation or repeated passes. This limitation is acute in dynamic zones: for instance, a narrow but deep drawdown at a glacier terminus or a channelized thinning on an ice shelf might be missed or poorly sampled by altimeter tracks. Furthermore, altimetry missions typically have longer revisited intervals (ICESat-2 repeats its ground track every 91 days) and require temporal averaging to create complete maps. In contrast, a single-pass SAR interferometer can map the area in near-real-time. The TanDEM-X mission already proved the value of this approach by mapping the entire Antarctic Ice Sheet at 12 m posting in 2013–2014(Wessel et al., 2021;Hajnsek et al., 2025). By avoiding temporal decorrelation, bistatic SAR captured even the steep and crevassed marginal areas that are challenging for repeat-track altimetry. For STV, which aims to monitor surface topography changes, leveraging such SAR capabilities means that large elevation changes, for example, the multi-meter thinning of a fast glacier or the dynamic rebound of an ice shelf after an iceberg calving, can be imaged as continuous surfaces rather than inferred from sparse points. Another advantage is that SAR can operate in all weather and lighting conditions; polar night or persistent cloud cover pose no obstacle for radar, whereas optical sensors (and to a lesser extent, lidar, which is impeded by clouds) have limitations in those regards. Our results also mitigate one of the traditional concerns with SAR, which is the penetration uncertainty.

We have shown that with appropriate correction, the X-band SAR elevations can be brought to parity with the true surface. This suggests that future missions can confidently exploit SAR interferometry for surface elevation, using bias correction models to reconcile SAR and optical measurements. In essence, the study supports a paradigm where SAR and lidar are complementary: SAR provides the 2D (or 3D) continuous elevation field, and lidar provides absolute elevation anchors on sparse tracks. This has direct relevance to the STV mission's cryosphere theme. Indeed, NASA's STV planning has recognized the value of combining measurements, for example, recent STV airborne campaigns have collected contemporaneous radar and lidar data to simulate such synergistic observations (Andrea & Craig, 2024). Our findings reinforce that strategy, demonstrating that the fusion of SAR interferometry with lidar ground-truth can yield high-accuracy, high-coverage elevation products. In particular, at ice sheet margins and grounding zones, where spatially continuous mapping is critical, SAR interferometry (especially in a single pass/bistatic mode to minimize decorrelation) stands out as the only practical way to




obtain wall-to-wall elevation change maps. Lidar altimetry, while extremely precise at a point scale (centimetre-level for ICESat-2), will remain a sampling system; it excels at capturing along-track profiles and small-scale features (like crevasse roughness or snow dune heights along the beam path but cannot seamlessly map large areas at once. Therefore, an STV mission

concept that combines the strengths of both, the continuity of SAR with the precision of lidar, is strongly supported by our results. We have effectively shown that the long-standing "radar penetration problem" can be overcome with AI-driven correction, meaning STV could confidently use an X- -band SAR to monitor ice surfaces, greatly expanding the mission's capability beyond what a lidar-only approach would achieve. This is especially pertinent for capturing extreme events or transient phenomena: for example, the rapid drawdown of an ice stream after an ice-shelf collapse, or the formation of a vast

melt lake lowering a region of an ice shelf. A SAR image pair could map the entire event, while a lidar satellite might only sample it.

While polar regions benefit from crossing orbits that naturally increase point density, mapping global topography with sub-meter accuracy at 50 m resolution using laser altimetry alone may require up to three years, largely due to limitations from cloud cover and revisit time (Degnan, 2016) . In contrast, bistatic InSAR offers a promising alternative for high-resolution

topographic monitoring. To evaluate its potential, we analyzed how elevation change accuracy improves as a function of acquisition frequency, assuming a linear surface change signal. Figure 14 illustrates the RMSE of elevation change estimates for a given DEM noise level relative to the number of acquisitions per year. Assuming a future InSAR mission replicating the current TanDEM-X formation, with an 11-day repeat cycle and year-round acquisition capability (similar to Sentinel-1 or NISAR) and using the DLR processor that achieves ~7 m global height accuracy (Rizzoli et al., 2017), we could reach sub-

meter accuracy in annual elevation change. Moreover, if our "InSAR-AI" bias correction and denoising approach (capable of achieving ~1 m single-measurement accuracy) can be generalized across different terrain types, the estimated accuracy of surface elevation change would improve to <0.6 m/year. By stacking three years of observations, this accuracy could further improve to ~10 cm/year (Figure 14). Such performance would approach that of laser altimetry in terms of vertical precision, but with the added advantage of less-accurate full spatial coverage and consistent acquisitions at every pass. This continuous

mapping capability is particularly beneficial for STV mission objectives, including monitoring vegetation structure, urban infrastructure, and other applications requiring seamless global topographic data. It is important to note, however, that this is a simplified analysis assuming random, normally distributed, and unbiased errors; potential effects of InSAR height of ambiguity are not accounted for here.

In summary, our demonstration of bias-corrected SAR elevations lends strong evidence that a SAR-equipped STV

mission would be able to map and monitor ice sheet changes with high fidelity. This represents a path forward for cryosphere remote sensing, where the trade-off between coverage and accuracy is no longer prohibitive and both can be achieved by smartly fusing technologies.





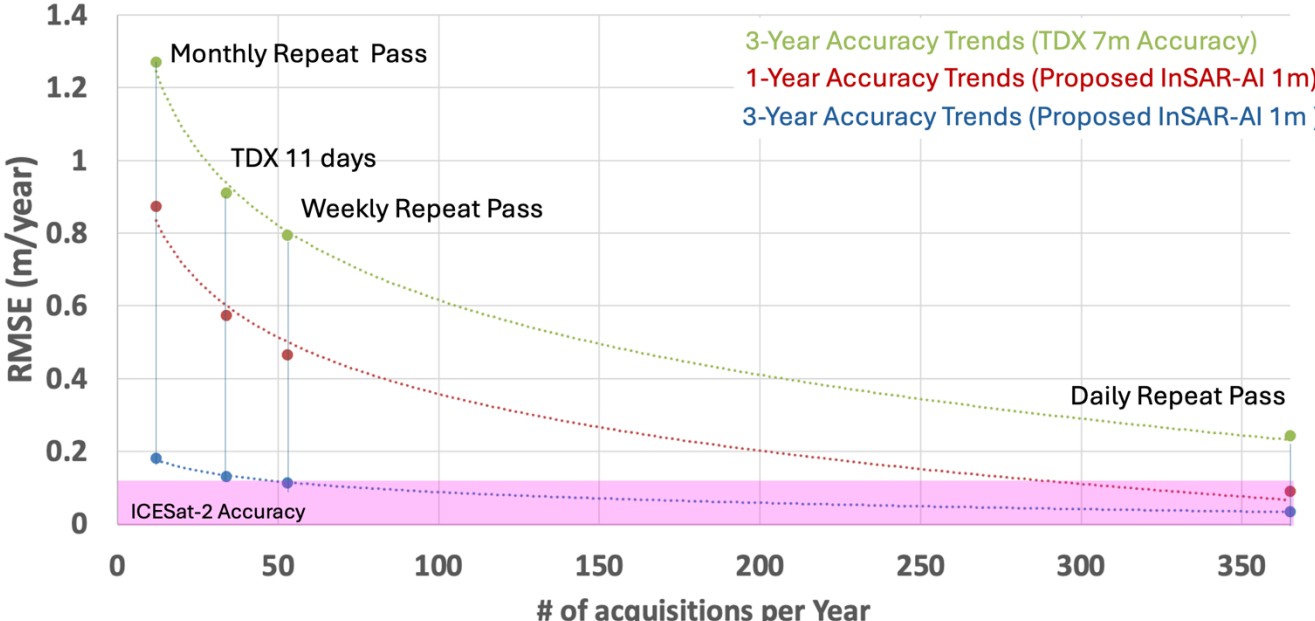

**Figure 14. RMSE elevation trend accuracy calculated for a given number of acquisitions per year. Bottom purple box indicates ICESat-2 accuracy of 10 cm.**

### 5.6 Proposed STV Operational Scenario: Lidar–SAR Data Fusion for Bias Correction

Building on the above, we propose an operational data fusion strategy for the cryospheric component of NASA's STV mission. The goal is to harness the synergy of lidar and SAR to continuously monitor ice elevations with minimal bias. A practical approach would be to implement a calibration phase at the start of the mission (and possibly at periodic intervals). During this phase, STV's lidar and SAR instruments would be deployed in a coordinated manner over representative ice regions, with data acquisitions closely timed. For instance, as shown in this research, over the course of 30 days, STV could target several key areas (e.g. East Antarctic plateau, a coastal outlet glacier, a West Antarctic ice shelf, and a mountain glacier region). In each area, the satellite (or satellite constellation) would acquire near-simultaneous lidar altimetry and bistatic SAR interferometry. By "near-simultaneous," we mean within a short window such that the surface change between acquisitions is negligible, our findings suggest within 15–30 days is sufficient to avoid significant physical changes, as we did with ICESat-2 and TanDEM-X. The purpose of these co-located measurements is to generate a training dataset where the true surface elevation is known from the lidar, and the corresponding biased SAR elevation (along with SAR observables like coherence, amplitude, and relevant meteorological data) is recorded.

This mirrors the methodology of our study, effectively creating thousands to millions of training points where the penetration bias can be computed as the difference between SAR and lidar heights. Once this calibration dataset is collected, a machine learning model would be trained to predict the penetration bias from the SAR and environmental inputs. In our case,





a neural network proved capable of learning the complex relationships, yielding bias predictions to ~1 m accuracy. STV could employ a similar or more advanced model, possibly retrained or fine-tuned for each region to capture regional variations in snow properties. The result of the training phase would be a set of model parameters that encode how to correct SAR elevations to match the lidar. After the calibration phase, the mission would transition to its operational monitoring phase. During this phase, the SAR instrument would continuously acquire interferometric measurements over the cryosphere to map surface elevation changes, and the trained model would be applied to these data to correct for penetration bias in real-time (or near-real-time). The lidar instrument, having provided ground-truth training data, could then be used more sparingly, for example, it might be tasked with other science targets (like vegetation), or used to occasionally validate the SAR-derived elevations at selected crossovers. Essentially, the lidar would no longer need to densely sample the ice sheet after calibration, because the SAR+ML system can fill in.

This represents a form of data fusion where the high precision of lidar is transferred to the SAR via the neural network correction. The benefits of this strategy are numerous. First, it maximizes coverage: the SAR can map large areas frequently (potentially every few days or weeks, depending on orbit and swath), providing continuous spatio-temporal coverage of ice elevation changes. Second, it preserves accuracy: the bias correction model ensures that each SAR-derived elevation is tied back to a lidar-calibrated truth, keeping errors on the order of decimetres to a meter at most. Third, it is efficient: instead of running two full missions independently, the lidar is used in a targeted way to bolster the SAR measurements. This concept is analogous to how Operation Ice Bridge under flew ICESat-2 to transfer the calibration between missions, but on a spaceborne scale and with machine learning in the loop. Importantly, the calibration is not a one-off, STV could repeat co-acquisitions periodically (for example, an annual or biannual check in various climates of Antarctica and Greenland) to update the model if needed.

The cryosphere is changing, and long-term climate shifts (like warming or changes in storminess) could alter penetration characteristics; a periodic recalibration ensures the model remains robust. Moreover, such a strategy de-risks the mission: even if the neural network approach under-performs in some area, the lidar can always directly measure those spots. Conversely, if the SAR operates flawlessly, one could even envision trading off some lidar coverage to conserve resources, since the network can propagate the calibration to unvisited areas as long as similar conditions were seen in the training. This approach aligns with the Decadal Survey's emphasis on multi-sensor integration and advanced algorithms. It essentially turns the STV mission into a fusion experiment where different observation modalities inform each other. As evidence that this is feasible, our study provides a proof of concept: using existing satellites, we successfully implemented the critical pieces (coincident data gathering, neural network training, bias correction, and validation). Implementing it operationally would institutionalize this synergy. The result would be a new level of capability in cryospheric monitoring: the mission could deliver continually updated, bias-corrected ice surface DEMs and time series, with the detail of SAR imagery and the accuracy of lidar. Such datasets would greatly benefit ice sheet models, sea-level projections, and hazard assessments (e.g. for ice shelf collapse or glacier runoff events). In conclusion, we recommend that STV consider a modus operandi wherein lidar and SAR are not redundant but complementary, with a dedicated cross-calibration period and real-time bias correction model. This



would exemplify a modern data fusion strategy, leveraging AI to combine sensor inputs. It is a pathway to overcome the historical trade-offs (coverage vs. accuracy) in remote sensing. This study's success in correcting X-band InSAR biases offers a template for such integration, underscoring that the technical hurdles can be overcome and that the synergy between lidar and SAR can be operationalized for the benefit of cryospheric science and sea-level rise research. (Donnellan & Glennie, 2024)

## 6 Conclusion

In this study, we developed a neural network-based approach to effectively predict and correct X-band radar penetration biases in TanDEM-X elevation data over Antarctica, achieving a RMSE accuracy of ~1 meter and a mean bias of 1 cm when validated against ICESat-2 altimetry. This substantial reduction in elevation uncertainty has significant implications for improving ice sheet digital elevation models, refining ice mass balance estimates, and enhancing melt rate calculations critical for climate and sea-level rise assessments. The integration of atmospheric variables from ECMWF provided an additional RMSE improvement of 10–20 cm (up to 15-20% improvement), underscoring the importance of temperature, wind speed, wind direction, and surface pressure in predicting radar penetration biases. Conversely, snow density showed minimal influence, likely due to coarse modelling of snowpack characteristics. SHAP analysis further revealed coherence and amplitude as dominant predictors, with their relative importance varying slightly between different acquisition intervals, possibly due to temporal variability and limited dataset size. Importantly, our findings validate SAR bistatic interferometry as a robust, spatially continuous method for mapping large-scale elevation changes, especially in grounding zones where lidar altimetry's discrete sampling approach is limited. We propose an operational scenario for NASA's STV mission that strategically combines lidar and bistatic SAR acquisitions during an initial calibration phase, enabling neural network training to operationally correct radar biases thereafter. Such a data fusion strategy would substantially enhance cryospheric monitoring capabilities, ultimately bridging the gap between high spatial coverage and elevation accuracy.

## Code and Data Availability

Elevation and ECMWF data and related code are available from the authors upon request and at https://github.com/Milillo-lab/TanDEM-X_ICESat-2_BiasEstimator.

## Acknowledgements

Author contributions: P.M. set up the Antarctica experiment and acquisition plans. P.R., and J.B.-B. processed the TDX time-tagged DEMs. J.N. Processed the ICESat-2 data. P.M., A.V. and A.C. interpreted the results and wrote the manuscript. All authors reviewed the manuscript. P.M. obtained research funding.

## Funding

This work was conducted at the University of Houston, TX under a contract with the Cryosphere Program of NASA. And the NASA Decadal Survey Incubation program.



**Data and materials availability**

We Thank NASA's ICESat-2 mission and the National Snow and Ice Data Center (NSIDC) for providing access to the data.
The TanDEM-X CoSSC products were provided by DLR through scientific proposal no. OTHER0103. All used CoSSCs can
be obtained for scientific purposes pending the submission of a scientific proposal, which should be submitted
to https://tandemx-science.dlr.de/. ECMW data were downloaded from the Copernicus Climate Change Service (Accessed
January 1st 2025).

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
