# Peer review of "AI-Driven TanDEM-X Penetration Bias Estimation in Antarctica Using ICESat-2 and ECMWF Data: Implications for the NASA Surface Topography and Vegetation Decadal Survey Incubation study"

_EGUsphere, 2025_

## Referee Comment (RC1)

Review of "AI-Driven TanDEM-X Penetration Bias Estimation in Antarctica Using ICESat-2 and ECMWF Data: Implications for the NASA Surface Topography and Vegetation Decadal Survey Incubation study" by Ankita Vashishtha et al.

Vashishtha et al. used a set of radar-derived parameters and a set of climate parameters as features in 14 non-linear regression models to predict the penetration bias of X-band microwave (defined as the difference between TanDEM-X heights and ICESat-2 heights). The radar-derived parameters and climate parameters are selected based on their data distributions and correlation coefficients between features. The most variable and independent parameters are adopted as features in the regression models. Based on the error metrics of the predicted penetration bias, the authors demonstrated the results of the RF and DNN models and their related sensitivity when the temporal difference between TanDEM-X and ICESat-2 acquisitions and the features in the regressors are altered. It was concluded from this analysis that incorporating the climate parameters into the regressors could improve the performance of the prediction, whereas the temporal difference between TanDEM-X and ICESat-2 acquisitions has a negligible impact. The authors also showed that InSAR coherence and amplitude are the most important features among the radar-derived parameters, while the 2 m air temperature, near-surface wind patterns and surface pressure are the most important features among the climate parameters that affect the regressors. Finally, implications for future missions and studies are proposed based on this study.

Overall, this study is insightful for the cryosphere community, as it provided a novel and interesting algorithm to derive high-resolution and potentially dynamic DEMs for Antarctica (perhaps also other snow-covered regions). The error metrics provided in the manuscript are also promising. The obtained X-band microwave penetration may also have the potential to assess subsurface snow/firn properties, which is interesting for cryosphere studies. However, my major concern is that the manuscript suffers from serious issues related to language, clarity, and structure. The current presentation lacks the level of professionalism expected in scientific writing, which makes it difficult to fully evaluate the contribution or impact of the work. A substantial rewrite is required to meet the standards expected for scientific publication. Please find the detailed comments below.

**General comments**
1. A lot of punctuation errors, subject-verb errors, errors in citation format, and informal languages are present throughout the manuscript (mainly up to Results section). The detailed examples will be provided in specific comments and technical corrections.

2. While I do not wish to discourage any innovative or fun ways of structuring a manuscript, I found the flow of the paper quite difficult to follow. In the Introduction section, the authors started by listing a wide range of satellite remote sensing techniques together with their limitations. Almost none of the listed techniques was used in this study. Suddenly, TanDEM-X was introduced, without any mention of the strengths and limitations. The regions of interest and data are proposed here, instead of conventionally in the Data section. An overview of the paper structure is proposed, followed by some literature reviews and (In)SAR principles as subsections...

Hereby I would like to suggest a preferred structure in the following order:
- Introduce the importance of accurately estimating (I would not say characterizing) Antarctic surface elevation and mass balance, as the authors already did.
- Introduce TanDEM-X, including what the mission is, what the mission does, and the strengths and limitations. The penetration bias and related previous studies can be introduced here as well.
- ICESat-2 data can be used to estimate penetration bias due to the theoretically negligible penetration into snow (Michel et al., 2014). However, the spatial gaps between the tracks, the cloud-cover impact, and the longer revisit time (compared to TanDEM-X) make it not ideal for constructing a high-resolution DEM solely based on ICESat-2 data. I understand that the authors already mentioned this limitation in the Discussion section. However, I would

recommend also introducing it here to clear the doubts such as "if the ICESat-2 data are already used as reference, why would the authors not use them to construct DEMs directly?"
- Introduce the opportunities brought by AI methods in estimating X-band penetration bias

In this way, Fig. 1 and Table 1 can also be moved to Study Area and Data section, in order to better match the titles of each section.

3. From Table 1, it is difficult to understand what the features and the target variable are. I would suggest adding a diagram or an explanation to clearly indicate what should be the input and output of the regression models.

4. I understand that it would be extra work and could be impossible to implement snow grain size in the regression models. However, I wonder what the authors think about the effect of snow grain size (in addition to density) on the estimation of penetration bias. As far as I understood, not only does snow grain size affect the volume scattering of radar (Tsang et al., 2022), but it can also introduce a scattering bias in ICESat-2 heights (Smith et al., 2018). I see that the effect of grain size is briefly mentioned in the Discussion section, but without a general discussion about what may affect the penetration bias, such a brief touch on the topic seems abrupt and unexplained.

5. I am curious about how the updated DEMs in the regions of interest compare with the existing CryoSat-2 DEM (Slater et al., 2018) and the Reference Elevation Model of Antarctica (REMA; Howat et al., 2019) in the corresponding regions. Would the difference between the DEMs indicate limitations in the proposed methodology, the CryoSat-2 methodology, or the stereoscopic-imagery-based methodology?

6. Line 183. "The typical winter season" is defined as the period between April and June. However, the Kuipers Munneke et al. (2018) study defined the winter season as the period between April and October. Could the authors motivate why they chose this period, and would it affect the result if the investigated time period is extended?

7. The authors put the assessment metrics of different regression models in the Supplement and only presented RF and DNN results in the main content, probably based on the best performance. Maybe I missed something, but the selection criteria and locations of the figures are not introduced very clearly. Table 5 is introduced in Line 370, but Fig. S1 only appeared in Line 374. This writing flow makes the readers wonder a bit why Table 5 only shows RF and DNN instead of the previously mentioned set of machine learning and deep learning models. I would suggest the authors add some comments about the result selection at the beginning of Section 4.1.

This is personal preference, but it might be clearer and simpler to move the figures from Supplement to Appendix.

**Specific comments**
1. Lines 19–28. This part of the Abstract seems somewhat repetitive, which dilutes the application of TanDEM-X and the importance of mitigating its penetration bias. I would suggest the authors change the flow into:

TanDEM-X is an InSAR mission operating in X-band (8–12 GHz frequency) -> it can be used to characterize snow and ice surfaces, analyze snow layers, and construct DEMs -> however, due to the absorption, reflection and transmission of X-band microwave in the snow layer, penetration biases are introduced and consequently cause the InSAR-derived DEMs to deviate from the true physical surface -> therefore, correcting X-band penetration biases is important...

I also wonder whether it is necessary at all to mention the characterization of snow and ice surfaces and the analysis of snow layers, because these applications are not mentioned afterwards anymore.

2. Lines 69–71. The listed features seem to cover both radar-derived parameters and topographic parameters. I would like to see the motivation for using them. Why not use more (or less) parameters than these?

Similarly, it would be more helpful to provide motivations for including the "environmental features" mentioned in Line 72.

3. Line 75. *"We then implemented five machine learning and nine deep learning algorithms, leveraging radar and environmental features to predict penetration bias values."*

I would suggest removing "values" for consistency in the manuscript.

4. Figure 1. The figure seems unnaturally flattened. Moreover, the map of Antarctica should typically be in polar stereographic projection visualized with an equal aspect ratio, so I would not recommend flattening it.

I wonder what the point of showing the IMBIE drainage basins is...the concept of drainage basins is not used in the rest of the manuscript.

Caption: "...for which ICESat-2 data, TanDEM-X data and ECMWF features is extracted" -> "...features are..."

5. Lines 99–104. Salinity is mentioned here multiple times, but is it useful for the application over the Antarctic Ice Sheet?

6. Line 106. *"InSAR coherence also effects the penetration depth..."*

This sentence is grammatically and logically wrong. I would recommend writing "a deeper penetration depth results in a lower InSAR coherence due to an increase in volume scattering (Deng et al., 2024)" or something similar.

7. Line 111. I am not sure that the cited work says *"overall backscatter intensity decreases due to surface scattering"*. What I found in Park et al. (2021) is: *"backscatter over wet snow can also be increased due to the complex wet snow metamorphism, including an increase in snow surface roughness and an increase in the snow grain size during overnight refreezing"*.

8. Lines 114–117. It is nice to distinguish the different concepts between "penetration depth" and "penetration bias", but it would be nicer to clarify again (apart from the title) that this study focuses on the bias.

9. Lines 126–131. Is there any reference for the concepts mentioned here? It would also be more concise and clear if the relationships could be expressed in the format of equations. For example, it seems that Eq. 9 of Dall (2007) is helpful.

10. Line 134. It would be nicer to briefly explain to the readers of The Cryosphere what "interferogram flattening" is.

11. Line 146. The penetration depths between 10 m and 12 m seem contradictory to what is mentioned below. These values also seem to deviate from Table 6 of Rizzoli et al. (2017).

12. Line 152. The Michel et al. (2014) work is not about InSAR penetration bias, but radar altimeter. Suggested removing.

13. Line 167. Please specify which "previous models" were applied for which purposes, and please also provide references.

14. Line 182. "These data have been collected between..." could be misunderstood, possibly indicating that ICESat-2 collects data only in winter seasons. I would recommend the authors be specific that they are the ones who selected these winter data.

15. Lines 189–194. I understand the authors would like to have a detailed motivation and background introduction about the time difference between TanDEM-X and ICESat-2. However, adding too much information before introducing the time difference adopted by the authors dilutes the real message. I would recommend that here it is sufficient to only keep Lines 195–199.

16. Line 209. *"To retrieve elevation profiles, the collected dataset of point elevation measurements is sampled by utilizing different beam spacings and arbitrary repeat-track geometries, along with the estimation of Root Mean Square (RMS) errors result into the elevation change measurements."*

The sentence does not seem optimal, both grammatically and logically. From what I understood, I would suggest rewriting into *"The elevation profiles are derived by sampling the collected dataset of point elevations using different beam spacing and arbitrary repeat-track geometries. The elevation change measurements are derived by computing the root mean square errors (RMSE)"*, or something similar. I am also not sure if I understood the differences (or similarities) between "elevation profiles" and "elevation change measurements". Hopefully the authors could clarify this.

In addition, throughout Section 2.1, I found it difficult to distinguish which part is the characteristics of ICESat-2 ATL06 product, and which part is post-processing of the authors. I would suggest the authors split the two processes in two separate paragraphs.

17. Line 227. *"TanDEM-X is also designed to maintain a sharp balance between height of ambiguity with phase disambiguation (Zink et al., 2021)."*

I wonder how important this balance is, as it does not seem to be mentioned in the result analysis. If it is really so important, I also recommend the authors introduce what phase disambiguation refers to to the readers of The Cryosphere.

18. Line 229–231. The data acquisition period is repetitive, as it was already mentioned in Line 182. Meanwhile, most of the acquisition periods in Table 2 are 2021 to 2023 instead of 2024. Please be consistent.

19. Lines 232–240. Is it possible to add a flowchart to show how ITP and TAXI work?

20. Line 249. *"Fourteen atmospheric features complement our dataset (Table 1)."*

Why would the authors use these fourteen features to "complement" the dataset? This introductory sentence is quite abrupt and does not really show the importance of using these features. Meanwhile, these features were called "environmental features" and now "atmospheric features". Please be consistent. Strictly speaking, snow density should also not be part of "atmospheric features".

As an additional comment on the Data section, I recommend clearly stating the purposes of each dataset used by the authors.

21. Line 259. I wonder what the added value of repeatedly mentioning (also the modifications made to) the CSV files throughout the Methodology section is. While I appreciate that the authors provided the code and introduced the correct data format to the potential users, I believe such specifications would be more appropriate in the code documentation. Given the already extensive length of the manuscript, I suggest omitting these minor details.

22. Figure 3. I would appreciate it if the input and output of the models could also be specified.

23. Lines 270–272. My concerns are the same as comment 21. The projection is not critical compared to the core logic and methodology of this manuscript.

24. Lines 279–282. *"Without any correction applied the differences between ICESat-2 elevation data and TanDEM-X DEM height are characterized by an RMSE of 4 m and median and mean value of -1.6 and -1.48 m along with minimum, maximum values as -18 m, 20 m respectively in line with previous studies (Rizzoli, et al., 2017)."*

This sentence here is unclear with a grammatical mistake. Suggested change: *"Without any correction applied, the differences between ICESat-2 heights and TanDEM-X heights are characterized by an RMSE of 4 m, a median of -1.6 m, a mean of -1.48 m, a minimum of -18 m, and a maximum of 20 m. The statistics are consistent with Rizzoli et al. (2017)."*

Please feel free to improve the idea based on this version. In general, I suggest the authors add commas between clauses and consider splitting long and detailed sentences to improve clarity.

25. Line 283. *"Firstly, we have removed the data points lying in the floating ice region by removing the data points having TanDEM-X DEM height less than 100m."*

I am curious why the authors are not interested in the floating-ice region. Moreover, this sentence could be more concise. I would suggest revising into *"Points with TanDEM-X DEM heights below 100 m were removed to exclude the floating-ice region."*

26. Line 286. What does "finite accuracy" mean? Why is it important for this study?

27. Line 287. *"For heights of ambiguity ranging between 30 and 80 meters expected tilts in a scene range between 7 to 19 cm (Gonzalez et al., 2024)."*

If I understood correctly, did the authors try to say *"For heights of ambiguity ranging between 30 and 80 meters, the expected tilts in a scene range between 7 cm to 19 cm (Gonzalez et al., 2024)"*? I am not sure if I understood the purpose of mentioning this range "7 cm to 19 cm". Is it somewhat connected to the previous sentence "low-varying offsets and tilts"? Does it work as a quality indicator? Is it useful for the setup of the following experiments? Please specify.

28. Line 289. If the points with a height of ambiguity less than 30 m only take up 1% of the dataset, how do they introduce imbalance? Should we consider them as outliers? What about the points with a height of ambiguity higher than 80 m? Why would the authors not apply the same outlier removal criteria shown in Eqs. 1 and 2 also for the height of ambiguity?

29. Lines 319–323. Many of the abbreviations do not seem to match the ones in Figs. S1–S4. I suggest double-checking.

30. Table 3. Could the authors explain how the hyperparameters listed in the table are chosen?

31. Line 335. I would avoid using words like "obviously", as they can sound somewhat casual, and the observation may not be as obvious to the reader as it is to the authors. This sentence has another problem. Please refer to item 11 of technical corrections.

32. Line 339. *"Based on this analysis we decide to remove some of the highly correlated variables and leave only seven features including temperature of the air at 2m above the land surface, snowfall, surface pressure, snow density, snow evaporation, wind speed and wind direction."* -> *"Based on this analysis, we removed several highly correlated variables and*

*kept the following seven features: 2 m air temperature, snowfall, surface pressure, snow density, snow evaporation, wind speed, and wind direction."*

But it seems like this selection is only applied to the so-called "environmental features". The following text gives the impression that the so-called "radar features" are all used. If this is the case, I suggest the authors state clearly what exactly the retained features are.

33. Figure 6 caption. *"ICESat-2 elevation data is not used as input; only TanDEM-X DEM values are considered for the AI models."*

Please check the consistency of the manuscript, as "data" should generally be plural and seems to be also used that way in most parts of this manuscript. Regarding this sentence itself, I find it appearing rather randomly. It does not really describe the figure, and the authors filtered out a lot of other parameters apart from the ICESat-2 heights, so why would the ICESat-2 elevation data be mentioned here only?

34. Line 361. I find it odd to suddenly mention passive sensors. They are not used in this study as far as I understood.

35. Line 366. This sentence is unclear due to grammatical issues and dense structure. If I understood correctly, it would be better to rewrite it into *"Initially, all models are trained using 6 radar features as input: TanDEM-X DEM height, coherence, amplitude, height of ambiguity, local incidence angle, and slope. Penetration bias is used as the target variable."* Please also decide whether the authors prefer to use TanDEM-X or TDX throughout the manuscript.

36. Line 368. This is another sentence that I am not sure how to interpret due to grammatical issues and dense structure. According to my understanding, I would suggest rewriting into *"For a 30-day time difference, the dataset contains 276,549 data points as input. This number is reduced to 215,620 when considering a 15-day time difference."* Please also check for consistency when you mention "time difference" or "time interval".

37. Line 383. "...RMSE values nearer..." I cannot see the point of using a comparative form of an adjective. Please rephrase.

It is also important to mention the units of the errors throughout the Results section.

38. Line 384. "The reason may be cited..." is quite unconventional. Based on my understanding, here is a suggested revision: *"The probable reason for the compromised performance of the Decision Tree regression model is that it uses a single tree, which makes decisions by splitting features at threshold values."* Please feel free to improve based on this, if I understood it wrong.

39. Line 393. The introduction of the features is repetitive. There should not be a comma after "including" in this sentence.

40. Line 395. This sentence is difficult to follow due to missing verbs, dense structure, and inconsistent pluralization. Please consider rewriting into *"Table 5 shows that DNN is the best-performing model among all the data models with an RMSE of approximately 1.04 m, an MAE of 0.74 m, an R2 score of 0.84, and an explained variance of 0.84",* or something similar. In general, please specify the reported metrics precisely and avoid vague phrases such as "almost equal to" or "along with high corresponding values".

41. Figs. 7–10. For a more straightforward comparison, I would suggest combining them into one figure. There could be four subplots showing four scatter-plots, and one subplot showing all histograms in different colors. In addition, these figures, together with Fig. 4, seem unnaturally stretched.

42. Line 401. "Similarly, Table 5..." Please refer to comment 40 to improve clarity.

43. Line 405. "Figure 9..." The sentence has the same problem as the comments mentioned above. Suggested rewriting: *"Figure 9 shows the DNN model results using data with a 15-day time difference. Compared to the case with a 30-day time difference (Figure 7), a slight improvement in performance is observed."* Please also check the consistency of the preferred word for "time difference between TanDEM-X and ICESat-2" throughout the manuscript.

44. Line 407. "...reduced between" is incorrect. Please specify: is it "reduced to" or "reduced by"?

45. Lines 410–419. This is a rather extensive description of the Supplementary material. As already mentioned in general comments, would it be clearer to put everything in Appendix?

46. Line 435. *"As the DNN model with model architecture given in Figure 11 is most efficient...a SHAP analysis..."* This sentence seems to imply that SHAP analysis was performed only because the DNN model showed the best performance. However, Line 459 mentions that SHAP analysis was also conducted for the RF model and the corresponding figures are not shown. This gives the impression that SHAP was applied to both models regardless of performance. To avoid confusion and to improve the flow, I recommend briefly introducing the SHAP analysis and its purpose in the Methods section, so that its use in the Results section does not seem unexplained.

47. Line 452. Would the result change if the authors use more or less than 1000 samples in the SHAP analysis?

48. Line 457. *"...wind speed, wind direction and surface pressure which also shows broad data distribution..."* Did the authors mean surface pressure shows a broad data distribution? But from what I understood, all input features should show a broad distribution. What is the purpose of mentioning it here?

49. Line 475. When the authors mentioned the corrected TanDEM-X DEM, is it possible to provide a map indicating the difference before and after the correction? I assume it can be interesting to see the spatial patterns of the improvement, apart from pure statistics.

50. Line 481. This was already mentioned in general comments. When the authors mentioned that the corrected TanDEM-X DEM achieves an accuracy similar to that of stereophotogrammetric DEMs, it would be interesting to see a comparison with REMA.

51. Line 482. It would be clearer to specify that the "block-wise calibration" is a procedure used in the Wessel et al. (2021) study. I am also not sure if I understood "residual elevation scatter". According to the Wessel et al. (2021) study, *"For the HPB the mean penetration bias and standard deviations vary from −1.68 to −5.66m and 0.92 to 1.20m, respectively. They are located in the interior of Antarctica and are well distributed over the continent to serve as ground control for the adjacent blocks."* These values (0.9 m to 1.2 m) seem to refer to the standard deviation of penetration bias.

52. Line 486. *"For example...DEM products can be integrated with other datasets..."* This sentence is extremely vague. Please specify how to integrated the DEM with which datasets for which purposes.

53. Line 490. Change detection is a similar idea to the "elevation change" concept mentioned in the following section. Would the authors consider merging the two sections and making the story more concise?

54. Line 498. *"This is critical because even modest systematic errors can translate into..."* It would be better to specify that these errors are in heights.

55. Line 509. *"Notably, in radar percolation zones where seasonal melt occurs, uncorrected InSAR measurements can differ by up to 10–15 m from the actual surface (Wessel et al., 2021)."*

I am not sure if I follow this logic. In principle, percolation zones with seasonal melt should be characterized by high-density rough layers on the surface and near the surface. I would not expect such a high penetration bias. I looked up how Wessel et al. (2021) commented on the difference between dry and percolation zones: *"Here, the temperatures are coldest (Macelloni et al., 2019; Scambos et al., 2018), and the SAR signal penetrates the most in dry, cold firn (Ulaby et al., 1986), whereas the coastal areas show lower penetration which clearly corresponds to the brighter reflecting percolation areas in the amplitude mosaic (Fig. 11). This variation in the SAR penetration over the whole AIS raises the absolute linear error (LE90) to 6.25m, which is calculated by sorting the absolute differences thresholded by 90% of the values."* I strongly recommend the authors specifying what they meant.

56. Line 514. "bias-corrected bistatic InSAR data": I assume the authors meant that they corrected for the DEM rather than the data themselves?

57. Lines 531–535. I am afraid I got lost in the logic of this paragraph. The paragraph seems to imply that over a long (30-day) time difference (again, please make it consistent and choose between difference, period and interval) between TanDEM-X and ICESat-2, both surface changes and volume changes can occur. But the authors seemed to try to explain why coherence outperforms amplitude, which means that the volume change should be more influential than surface changes?

58. Line 546. *"...underscoring that encapsulates a different aspect of the scattering physics..."* I tried to help revise this, but this part of the sentence is hardly understandable.

59. Line 587–589. I am not sure if I understood it. Did the authors mean that the ECMWF data are in a coarse spatial resolution, whereas the real-world climate effects are rather local hence require a high-resolution model?

60. Line 610. *"...confirms that SAR can serve as a powerful tool for measuring ice topography..."* Would it be more precise to say it is bistatic InSAR? Furthermore, on Line 615, the concept of "swath-based InSAR" is suddenly introduced. I would strongly recommend the authors be more consistent with the language they use to refer to TanDEM-X.

61. Line 620. "deep drawdown at a glacier terminus": does it mean a steep topography or a pronounced ice loss?

62. Line 630. I wonder why optical sensors should be mentioned suddenly.

**Technical corrections**
1. Please check in Table 1 and throughout the manuscript the misspelling of ECMWF.

2. Line 133. "...and interferometric phase change..." -> "...an..."

3. Line 134 and throughout the manuscript. Please check that there should be a space between the text and the citation. Some commas and dots are missed here and there.

4. Line 135. "almost equals" -> "almost equal"

5. There are two Rizzoli et al. (2017) reference entries that should be distinguished with 2017a and 2017b.

6. Line 164 and throughout the manuscript. The use of "like" to introduce examples is informal in academic writing. It is strongly recommended replacing them with "such as".

7. Line 197 and throughout the manuscript. The abbreviation TDX is used in parallel with TanDEM-X. I would suggest the authors remain consistent with only one expression.

8. Line 205. "...with each pair is separated by distance of 3km..." -> "...with each pair separated by 3 km..."

9. Line 207 and many other places in the manuscript. (B. Smith et al., 2019) -> (Smith et al., 2019)

10. Line 214. "signal-to-noise ratio"

11. Line 215 and throughout the manuscript. The present continuous tense is generally not considered appropriate in academic writing. The accepted tenses should be past simple, present simple and present perfect. It is strongly recommended thoroughly checking it throughout the manuscript.

12. Line 273. "...the corresponding TanDEM-X elevation data and associated radar features is extracted from..." -> "...are extracted from..."

13. Line 284. "We consider only points where the TanDEM-X coherence is larger than 0.3 as high coherence values lead to a more accurate estimate of the interferometric phase." -> "We consider only points where the TanDEM-X coherence is larger than 0.3, as high coherence values lead to a more accurate estimate of the interferometric phase."

14. Line 285 and throughout the manuscript. When a reference entry is mentioned within the texts, it should be cited as "Gonzalez et al. (2024)". I recommend the authors double check the citation rules.

Also this line, it should be "Similarly to Gonzalez et al. (2024), we..." For similar problems, please refer to item 24 of specific comments.

15. Line 311. "Before data feeding in the machine learning and deep learning algorithms, data is divided into training, validation and testing data." -> "Before being fed into the machine learning and deep learning algorithms, the data is divided into training, validation, and testing sets," or "Before feeding data into the machine learning and deep learning algorithms, we divide the data into training, validation, and testing sets."

16. Line 323. "The accuracy of all these machine learning and neural network models are compared..." -> "...is compared..."

17. Line 334. "To understand potential interrelations between radar and atmospheric features we focus to plot histograms of our variables and calculate their respective correlation (Figure 5, Figure 6)." -> "To understand potential interrelations between radar and atmospheric features, we focus on plotting histograms of our variables and calculating their respective correlation (Figs. 5–6)."

18. Line 336. "Various environmental features like Ultraviolet Albedo and Near Infrared Albedo along with snow depth show high values of correlation." -> "Various environmental features, such as ultraviolet albedo, near-infrared albedo, and snow depth, show high values of correlation."

19. Line 337 and multiple places in the manuscript. This is an incorrect use of "whereas"; it is not typically used to start a sentence.

20. Line 344. "...ICESat-2 elevation data is excluded" -> "...data are..." Please also check throughout the manuscript for consistency.

21. Line 376. "best performed" -> "best-performing"

22. Line 386. "However, it's not much effective when there is a complex interaction between the features of the input data. Whereas Random Forest which is an ensemble of multiple decision trees is very effective in complex non-linear regression tasks."

This paragraph contains serious issues in both grammar and clarity. The first sentence is grammatically incorrect ("it's not much effective" is not standard English and should be avoided in academic writing). The second sentence should not start with "whereas", and the contrast introduced by "whereas" is misused (it is difficult to understand what kind of comparison is achieved here). I strongly recommend rewriting this part to improve both clarity and readability.

23. Line 399. "...a gaussian distribution of errors suggesting that..." -> "a Gaussian distribution of errors, suggesting that..."

24. Line 404. A comma should be added after "Interestingly".

25. Line 454. Suggested revision: "The top five features which influence the DNN model prediction ability include coherence, amplitude, height of ambiguity, TanDEM-X DEM height values, and 2 m temperature. These features are ranked by importance."

26. Line 456. "effecting" -> "affecting" or "influential"

27. Line 456. "Other significant atmospheric features which are affecting the DNN model includes wind speed..." -> "Other significant atmospheric features affecting the DNN model include wind speed..."

In this case, one can use a present participle ("affecting"), as long as it serves as part of a reduced relative clause.

28. Line 457. Suggested revision: "For 30-day-difference data, the top five features..."

29. Line 458. "The top performing features is coherence followed by amplitude. " -> "The best-performing features are coherence and amplitude."

30. Line 459. "...both types of datasets"

31. Line 459. Suggested revision: "We also conducted a SHAP analysis for the Random Forest model, the second-best-performing regression model after DNN. The SHAP analysis of the Random Forest model gave the same result as the DNN model for both the 15- and 30-day cases."

32. Line 475. "...height achieves..."

33. Line 477. "about" is rather informal. I would change it into "approximately".

34. Line 547. "...coherence excels when capturing volume scattering, while amplitude captures persistent surface characteristics, they are both essential." -> "...coherence excels when capturing volume scattering, while amplitude captures persistent surface characteristics; they are both essential."

35. Line 598. "Finally, the low importance of snow density underscores the need for improved snow/firn modeling in polar regions, current models may not represent the features most relevant to X-band radar, such as layering or grain size, reinforcing the value of using observable proxies like temperature and wind in the meantime." -> "Finally, the low importance of snow density underscores the need for improved snow and firn modeling in polar regions. Current models may not represent the features most relevant to X-band radar,

such as layering or grain size, reinforcing the value of using observable proxies such as temperature and wind."

**Reference**
Dall, J.: InSAR elevation bias caused by penetration into uniform volumes, IEEE Transactions on Geoscience and Remote Sensing, 45(7), 2319–2324, https://doi.org/10.1109/TGRS.2007.896613, 2007.

Howat, I. M., Porter, C., Smith, B. E., Noh, M.-J., and Morin, P.: The Reference Elevation Model of Antarctica, The Cryosphere, 13, 665–674, https://doi.org/10.5194/tc-13-665-2019, 2019.

Kuipers Munneke, P., Luckman, A. J., Bevan, S. L., Smeets, C. J. P. P., Gilbert, E., van den Broeke, M. R., Wang, W., Zender, C., Hubbard, B., Ashmore, D., Orr, A., and King, J. C.: Intense Winter Surface Melt on an Antarctic Ice Shelf, Geophys. Res. Lett., 45, 7615–7623, https://doi.org/10.1029/2018gl077899, 2018.

Michel, A., Flament, T., and Rémy, F.: Study of the Penetration Bias of ENVISAT Altimeter Observations over Antarctica in Comparison to ICESat Observations, Remote Sensing, 6, 9412–9434, https://doi.org/10.3390/rs6109412, 2014.

Park, J., Forman, B. A., & Lievens, H.: Prediction of Active Microwave Backscatter over Snow-Covered Terrain across Western Colorado Using a Land Surface Model and Support Vector Machine Regression, IEEE Journal of Selected Topics in Applied Earth Observations and Remote Sensing, 14, 2403–2417, https://doi.org/10.1109/JSTARS.2021.3053945, 2021.

Rizzoli, P., Martone, M., Rott, H., & Moreira, A.: Characterization of snow facies on the Greenland ice sheet observed by TanDEM-X interferometric SAR data, Remote Sensing, 9(4),1-24, https://doi.org/10.3390/rs9040315, 2017.

Slater, T., Shepherd, A., McMillan, M., Muir, A., Gilbert, L., Hogg, A. E., Konrad, H., and Parrinello, T.: A new digital elevation model of Antarctica derived from CryoSat-2 altimetry, The Cryosphere, 12, 1551–1562, https://doi.org/10.5194/tc-12-1551-2018, 2018.

Smith, B. E., Gardner, A., Schneider, A., and Flanner, M.: Modeling biases in laser-altimetry measurements caused by scattering of green light in snow, Remote Sensing of Environment, 215, 398–410, https://doi.org/10.1016/j.rse.2018.06.012, 2018.

Tsang, L., Durand, M., Derksen, C., Barros, A. P., Kang, D.-H., Lievens, H., Marshall, H.-P., Zhu, J., Johnson, J., King, J., Lemmetyinen, J., Sandells, M., Rutter, N., Siqueira, P., Nolin, A., Osmanoglu, B., Vuyovich, C., Kim, E., Taylor, D., Merkouriadi, I., Brucker, L., Navari, M., Dumont, M., Kelly, R., Kim, R. S., Liao, T.-H., Borah, F., and Xu, X.: Review article: Global monitoring of snow water equivalent using high-frequency radar remote sensing, The Cryosphere, 16, 3531–3573, https://doi.org/10.5194/tc-16-3531-2022, 2022.

Wessel, B., Huber, M., Wohlfart, C., Bertram, A., Osterkamp, N., Marschalk, U., Gruber, A., Reuß, F., Abdullahi, S., Georg, I., and Roth, A.: TanDEM-X PolarDEM 90m of Antarctica: generation and error characterization, The Cryosphere, 15, 5241–5260, https://doi.org/10.5194/tc-15-5241-2021, 2021.

---

## Author Comment (AC1)

**Comment submitted by Reviewer-2 on 21/08/2025**

**Comments: Thank you to the authors for this contribution. The work is novel and will be impactful for developing multisensor fusion architectures under the Surface Topography and Vegetation mission. After looking closely at the previous review for this submission, I do not have any further comments; I agree with the previous reviewer in that I expect a more professional written presentation before initial manuscript submission. Please revise this manuscript following the suggestions given by the previous reviewer. This will allow me to provide a more focused scientific review for the next round of revisions.**

Remarks: We sincerely thank the reviewer for their positive feedback and for identifying the novelty and potential impact of our work for multisensory fusion architectures under the Surface Topography and Vegetation mission. We agree with the reviewer's suggestion regarding the written presentation of the manuscript. Following the comments provided by the first reviewer, we have made the requisite corrections in the manuscript to present more professional writing.

---

## Author Comment (AC2)

**Community Comment:**

**Dear authors,**
**very interesting approach! This is just one question/comment from selectively reading the preprint.**

**Whereas the relation between penetration bias and radar parameters (coherence, amplitude) is well know and understood (which makes it powerful for AI-based penetration bias corrections, such as yours), I find it a clever idea to include environmental parameters for an additional performance gain. As far as I understand, you use only current environmental parameters within 15 or 30 days of the TanDEM-X acquisition. This has a clear relevance, for instance about snow wetness, as you describe and discuss.**

**However, since the penetration is clearly related to the firn properties below the surface (e.g. stratigraphy, presence of refrozen melt layers, grain sizes, ...), I'm wondering if the environmental parameters of the recent few years might be actually more relevant than only the current ones within 30 days of the SAR acquisition. The environmental parameters of the past few years could be a good proxy for the subsurface firn structure/properties that determine signal penetration.**
**What's your take on this? Did you explore using environmental parameters from the previous years? An implementation of this probably triggers a couple of further questions, so I guess this might be something for future research. I still would be interested to hear a comment about this from you.**

**Best regards,**
**Georg Fischer**

Reply: Thank you, Dr. Georg Fischer, for your thoughtful comments and valuable feedback on our preprint. The penetration depth of SAR signals is highly dependent on the current snow surface properties, which are known to be highly dynamic. We have not tested the impact of environmental parameters in the past few years, but we reserve to work on this in future studies.

---

## Author Comment (AC3)

Comments submitted by Reviewer-1 on 29/06/2025

**Review of "AI-Driven TanDEM-X Penetration Bias Estimation in Antarctica Using ICESat-2 and ECMWF Data: Implications for the NASA Surface Topography and Vegetation Decadal Survey Incubation study" by Ankita Vashishtha et al.**

**Vashishtha et al. used a set of radar-derived parameters and a set of climate parameters as features in 14 non-linear regression models to predict the penetration bias of X-band microwave (defined as the difference between TanDEM-X heights and ICESat-2 heights). The radar-derived parameters and climate parameters are selected based on their data distributions and correlation coefficients between features. The most variable and independent parameters are adopted as features in the regression models. Based on the error metrics of the predicted penetration bias, the authors demonstrated the results of the RF and DNN models and their related sensitivity when the temporal difference between TanDEM-X and ICESat-2 acquisitions and the features in the regressors are altered. It was concluded from this analysis that incorporating the climate parameters into the regressors could improve the performance of the prediction, whereas the temporal difference between TanDEM-X and ICESat-2 acquisitions has a negligible impact. The authors also showed that InSAR coherence and amplitude are the most important features among the radar-derived parameters, while the 2 m air temperature, near-surface wind patterns and surface pressure are the most important features among the climate parameters that affect the regressors. Finally, implications for future missions and studies are proposed based on this study.**

**Overall, this study is insightful for the cryosphere community, as it provided a novel and interesting algorithm to derive high-resolution and potentially dynamic DEMs for Antarctica (perhaps also other snow-covered regions). The error metrics provided in the manuscript are also promising. The obtained X-band microwave penetration may also have the potential to assess subsurface snow/firn properties, which is interesting for cryosphere studies. However, my major concern is that the manuscript suffers from serious issues related to language, clarity, and structure. The current presentation lacks the level of professionalism expected in scientific writing, which makes it difficult to fully evaluate the contribution or impact of the work. A substantial rewrite is required to meet the standards expected for scientific publication. Please find the detailed comments below.**

**General comments**

**1. A lot of punctuation errors, subject-verb errors, errors in citation format, and informal languages are present throughout the manuscript (mainly up to Results section). The detailed examples will be provided in specific comments and technical corrections.**

Thank you for your valuable comments. All the punctuation errors, subject-verb errors, errors in citation format, and informal language corrections have been corrected in the new version of the manuscript.

**2. While I do not wish to discourage any innovative or fun ways of structuring a manuscript, I found the flow of the paper quite difficult to follow. In the Introduction section, the authors started by listing a wide range of satellite remote sensing techniques together with their limitations. Almost none of the listed techniques was used in this study. Suddenly, TanDEM-X was introduced, without any mention of the strengths and limitations. The regions of interest and data are proposed here, instead of conventionally in the Data section. An overview of the paper structure is proposed, followed by some literature reviews and (In)SAR principles as subsections...**

**Hereby I would like to suggest a preferred structure in the following order:**
**- Introduce the importance of accurately estimating (I would not say characterizing) Antarctic surface elevation and mass balance, as the authors already did.**

**- Introduce TanDEM-X, including what the mission is, what the mission does, and the strengths and limitations. The penetration bias and related previous studies can be introduced here as well.**

**- ICESat-2 data can be used to estimate penetration bias due to the theoretically negligible penetration into snow (Michel et al., 2014). However, the spatial gaps between the tracks, the cloud-cover impact, and the longer revisit time (compared to TanDEM-X) make it not ideal for constructing a high-resolution DEM solely based on ICESat-2 data. I understand that the authors already mentioned this limitation in the Discussion section. However, I would recommend also introducing it here to clear the doubts such as "if the ICESat-2 data are already used as reference, why would the authors not use them to construct DEMs directly?"**

**- Introduce the opportunities brought by AI methods in estimating X-band penetration bias.**

**In this way, Fig. 1 and Table 1 can also be moved to Study Area and Data section, in order to better match the titles of each section.**

Agreed, we have revised the introduction of our paper accordingly on page nos.2-3, line nos.41-82:

Accurately estimating Antarctic surface elevation and mass balance is critical for understanding polar climate dynamics and projecting future sea-level rise (Pritchard et al., 2009; Flament & Rémy, 2012; Paolo et al., 2023; Fricker et al., 2025). The Antarctic Ice Sheet spans ~14.2 million km², making satellite remote sensing the only practical approach for sustained, continent-scale monitoring. Surface elevation change is a key observable for determining ice mass balance, and precise measurements are essential to detect subtle signals of ice dynamics, accumulation, and melt.

Among available remote sensing techniques, TDX bistatic synthetic aperture radar (SAR) interferometry offers high-resolution, spatially continuous digital elevation models (DEMs) at X-band (wavelength ~3 cm). The mission, comprising the TerraSAR-X and TDX satellites flying in close formation, acquires single-pass interferometric data that avoids temporal decorrelation and allows mapping of steep and complex Antarctic terrains (Krieger et al., 2013; Zink et al., 2021). TDX is also designed to maintain a sharp balance between the height of ambiguity with phase

disambiguation (Zink et al., 2021). Phase disambiguation, also known as phase unwrapping, is the process of resolving the modulo-$2\pi$ ambiguity integrated in interferometric phase measurements to retrieve absolute surface displacement or elevation differences from wrapped interferometric data. However, X-band radar signals penetrate dry snow and firn, leading to penetration bias, where the interferometric phase center lies below the actual physical surface (Rizzoli et al., 2017b; Rott et al., 2021). The magnitude of this bias depends on snow and ice properties (density, grain size, liquid water content), acquisition geometry, and environmental conditions, and can reach several meters in cold, dry interiors (Page & Ramseier, 1975; Guo et al., 2023; Nandan et al., 2017; Tsang et al., 2022; Abdullahi et al., 2024; Park et al., 2021; Achammer & Denoth, 1994; Leinss et al., 2014). Understanding and correcting this bias is essential for integrating TDX data with non-penetrating sensors and for obtaining true surface elevations.

A natural reference for quantifying X-band penetration bias is provided by ICESat-2 laser altimetry, which operates at 532 nm and has negligible penetration into snow and ice (Michel et al., 2014). ICESat-2 delivers centimeter-scale elevation accuracy (Brunt et al., 2019) but, as a profiling instrument, it suffers from spatial gaps between ground tracks, susceptibility to cloud cover, and a 91-day repeat cycle. These factors make it unsuitable for constructing complete, high-resolution DEMs on its own, especially in non-polar dynamic regions or during short observation windows. Instead, ICESat-2's strength lies in serving as a precise but spatially sparse "truth" dataset for calibrating and validating radar-derived DEMs.

The ability to correct TDX penetration bias using ICESat-2 has been explored in several studies, but traditional methods often rely on limited regions or a small set of predictors (e.g., coherence, backscatter intensity) (Hoen & Zebker, 2000; Stebler et al., 2005; Dall, 2007; Oveisgharan & Zebker, 2007; Abdullahi et al., 2019; Fischer et al., 2019; Fischer et al., 2020; Li et al., 2021; Park et al., 2021; Rott et al., 2021; Abdullahi et al., 2024; Bannwart et al., 2024). Recent advances in artificial intelligence (AI) and machine learning offer the opportunity to model this bias more comprehensively, using both SAR observables and auxiliary environmental variables (e.g., snow depth, temperature, wind speed) to capture complex, nonlinear relationships.

In this study, we develop and test AI-based models for predicting and correcting TDX penetration bias across diverse Antarctic environments. By combining more than 300,000 near-coincident TDX and ICESat-2 measurements (within 30 days) with environmental parameters from ECMWF ERA-5 reanalysis, we aim to produce bias-corrected DEMs with ~1 m RMSE relative to ICESat-2. Our approach is applied to multiple regions, including grounding zones of major East Antarctic glaciers (Amery, Cook, Denman, Reid, Totten, Moscow University, Holmes) and selected areas in Queen Maud Land and the Weddell Sea.

By integrating the complementary strengths of TDX and ICESat-2 with machine learning, this work aims to bridge the gap between sparse, high-precision laser altimetry and spatially continuous radar interferometry. Our results provide a framework for generating more accurate, bias-corrected DEMs, supporting improved assessments of Antarctic mass balance and its contribution to global sea-level change.

**3. From Table 1, it is difficult to understand what the features and the target variable are. I would suggest adding a diagram or an explanation to clearly indicate what should be the input and output of the regression models.**

Thank you for your comment. In the new version of the manuscript, data input and output of the AI models are mentioned in the methodology and results sections. Additionally, we have also added the explanation below Table 1 on page no.10, line nos. 242-246 given below:

For the ICESat-2 datapoints, all the parameters are extracted as explained in Table 1. However, for input into the machine learning and deep learning models, we used six radar features, namely TDX DEM height, coherence, amplitude, height of ambiguity, local incidence angle, and slope as well as seven environmental features, including temperature of the air at 2 m above the land surface, snowfall, surface pressure, snow density, snow evaporation, wind speed, and wind direction to predict penetration bias as the output.

**4. I understand that it would be extra work and could be impossible to implement snow grain size in the regression models. However, I wonder what the authors think about the effect of snow grain size (in addition to density) on the estimation of penetration bias. As far as I understood, not only does snow grain size affect the volume scattering of radar (Tsang et al., 2022), but it can also introduce a scattering bias in ICESat-2 heights (Smith et al., 2018). I see that the effect of grain size is briefly mentioned in the Discussion section, but without a general discussion about what may affect the penetration bias, such a brief touch on the topic seems abrupt and unexplained.**

Thank you for your valuable suggestions. To explain the effect of snow grain size on the penetration bias, we have added an explanation regarding the effect of snow grain on penetration bias in the introduction part at page no.3, line nos.93-97, as given below:

Variations in snow water equivalent (SWE) significantly influence the physical properties of snow, such as snow grain size, which tends to decrease as SWE increases(Tsang et al., 2022). However, radar backscatter increases with SWE and grain size, while both factors are inversely related to penetration bias. Larger grain size enhances volume scattering within the snowpack, thereby limiting radar penetration and reducing penetration bias (Tsang et al., 2022).

Further, we acknowledge that snow grain size influences penetration bias estimation; however, since this parameter is not provided in the ECMWF dataset, generating it would require a separate model, which falls beyond the scope of this work. So, instead of snow grain, we have considered the parameter of snow density, which is available in the ECMWF data. The Antarctica region consists of a structured snowpack with highly varying snow grain radii and densities with depth (Brucker et al., 2010). As such, there is no direct relationship between snow density and grain size; however, the same can be linked with the formula of specific surface area (SSA) of a snow region given by the formula (Warren, 2019),

$$SSA = \frac{3}{r_e \rho_{ice}}$$

Where, $\rho_{ice}$ is the ice density and $r_e$ is the radius of the snow particle.

From the SSA formula, snow density and grain size are not directly linked in the equation itself; however, their mutual dependence on SSA indicates an indirect proportionality, where larger grain sizes are often associated with higher densities. In our work, as snow grain size data is unavailable, we have used snow density data as an input to the models for predicting penetration bias. As given in Figures 12 and 13, SHAP analysis shows that snow density is the least affecting parameter on penetration bias estimation. As grain size is proportional to snow density, including grain size as input to models may not affect the penetration bias prediction.

**5. I am curious about how the updated DEMs in the regions of interest compare with the existing CryoSat-2 DEM (Slater et al., 2018) and the Reference Elevation Model of Antarctica (REMA; Howat et al., 2019) in the corresponding regions. Would the difference between the DEMs indicate limitations in the proposed methodology, the CryoSat-2 methodology, or the stereoscopic-imagery-based methodology?**

Thank you for your question. This study aims to estimate the penetration bias in time-tagged TanDEM-X DEMs (acquired at 12 m spatial resolution), for acquisitions between 2021 and 2024. In contrast, the CryoSat-2 and REMA DEMs represent temporally averaged elevation products from CryoSat-2 from radar altimetry (2010–2016) and REMA from optical imagery (2009–2017) and do not provide timestamps for individual acquisitions. Their distinct temporal coverage and resolutions, considering the high dynamic environments at the grounding lines we analyze here, make direct comparisons with our compensated TanDEM-X DEMs unsuitable for evaluating method performance for any of the three sensors. We show below a plot in Figure 1, comparing our TanDEM-X DEMs with REMA, showing how these two products differ in several areas of our domain.

[Figure]

**Figure 1. Comparison of the TanDEM-X bias-corrected DEM with the REMA DEM. (i) Map of Antarctica showing the four selected locations (A-D), (ii)DEM comparisons for each location are shown in corresponding rows, where DEM height difference is calculated as TanDEM-X bias corrected-REMA DEM, and their histograms are also plotted.**

**6. Line 183. "The typical winter season" is defined as the period between April and June. However, the Kuipers Munneke et al. (2018) study defined the winter season as the period between April and October. Could the authors motivate why they chose this period, and would it affect the result if the investigated time period is extended?**

Thank you for your observation. The period between April and June is selected as per the availability of ICESat-2 and TanDEM-X datasets for the 8 regions in the East Antarctica region (Figure 2) (Table 2). We have clarified this in the current version of the manuscript.

**7. The authors put the assessment metrics of different regression models in the Supplement and only presented RF and DNN results in the main content, probably based on the best performance. Maybe I missed something, but the selection criteria and locations of the figures are not introduced very clearly. Table 5 is introduced in Line 370, but Fig. S1 only appeared in Line 374. This writing flow makes the readers wonder a bit why Table 5 only shows RF and DNN instead of the previously mentioned set of machine learning and deep learning models. I would suggest the authors add some comments about the result selection at the beginning of Section 4.1.**

**This is personal preference, but it might be clearer and simpler to move the figures from Supplement to Appendix.**

Thank you for your suggestion. We have changed the arrangement of the regression orders based on the Associate Editor's comments during the manuscript submission phase. We accommodate this request now and explain this better at the beginning of Section 4.1 and by shifting figures S1 to S4 in the annexure on page nos.38-41.

**Specific comments**

**1. Lines 19–28. This part of the Abstract seems somewhat repetitive, which dilutes the application of TanDEM-X and the importance of mitigating its penetration bias. I would suggest the authors change the flow into:**

**TanDEM-X is an InSAR mission operating in X-band (8–12 GHz frequency) -> it can be used to characterize snow and ice surfaces, analyze snow layers, and construct DEMs -> however, due to the absorption, reflection and transmission of X-band microwave in the snow layer, penetration biases are introduced and consequently cause the InSAR-derived DEMs to deviate from the true physical surface -> therefore, correcting X-band penetration biases is important...**

**I also wonder whether it is necessary at all to mention the characterization of snow and ice surfaces and the analysis of snow layers, because these applications are not mentioned afterwards anymore.**

Thank you for your valuable suggestions. As recommended, we have revised the abstract at page nos.1-2, line nos.19-38, to improve its flow as given below:

TanDEM-X is an InSAR mission operating in X-band (8–12 GHz frequency). It can be used to characterize snow and ice surfaces, analyze snow layers, and construct DEMs. However, due to the absorption, reflection, and transmission of X-band microwave in the snow layer, penetration biases are introduced and consequently cause the InSAR-derived DEMs to deviate from the true physical surface, affecting the accuracy of glacial mass balance estimation. This bias complicates efforts to integrate SAR-derived digital elevation models (DEMs) with laser altimetry data such as ICESat-2, where penetration is negligible. Therefore, correcting X-band biases is critical for achieving high-accuracy surface elevation models and improving glaciological interpretations. Here, we address these challenges by integrating neural network techniques with TanDEM-X (TDX) DEMs, Ice, Cloud, and land Elevation Satellite-2 (ICESat-2) altimetry data, and environmental parameters from the European Centre for Medium-Range Weather Forecasts (ECMWF). We leverage about 300,000 ICESat-2 pointwise measurements acquired within 30 days of TDX measurements spanning 2021-2024 in Antarctica. Additionally, we consider a diverse dataset of snow and atmospheric variables, including temperature, snowfall, snow depth, and wind speed, to model and predict X-band penetration biases across Antarctica. This approach advances existing methods by automating bias correction and enhancing the integration of SAR and altimetry datasets, achieving a mean bias correction of the order 1 cm with a Root Mean Square Error (RMSE) of about 1 m and maximum errors of the order of 10 m. Findings from this work provide actionable insights for improving elevation model accuracy in different ways. First, we offer the retrieved pointwise TDX, ICESat-2, and ECMWF datasets open access for future studies through a dedicated GitHub page. Second, we provide both our trained network weights and our algorithms in the form of Jupyter Notebooks for improved reproducibility. Most importantly, we discuss the broader efforts in the NASA Surface Topography Elevation (STV) decadal survey incubation study and polar ice monitoring by addressing penetration biases, which is one of the most critical uncertainties in radar remote sensing of snow and ice.

**2. Lines 69–71. The listed features seem to cover both radar-derived parameters and topographic parameters. I would like to see the motivation for using them. Why not use more (or less) parameters than these?**

**Similarly, it would be more helpful to provide motivations for including the "environmental features" mentioned in Line 72.**

Thank you for your comments. We have used radar features such as TanDEM-X DEM height, amplitude, height of ambiguity, coherence, local incidence angle, and surface slope for model input, as these features collectively characterize the physical and geometric conditions affecting X-band radar signals' penetration and phase integrity. Further, we have also extracted 14 environmental features from the ECMWF dataset, which are selected based on literature indicating their influence on X-band radar signal penetration in snow and firn, which affect snow density, grain size, and moisture content. Further, we leave out some parameters as they do not contribute to the prediction because highly correlated with other ECWF parameters. We do not use more parameters as these are the only available from ECMWF, and we would like to keep our algorithm as generalizable as possible. We have now remodeled the introduction as requested in the previous comment and explained this in the manuscript. We have also added a detailed explanation about selecting environmental features on page no.12, line no.292-302.

**3. Line 75. "We then implemented five machine learning and nine deep learning algorithms, leveraging radar and environmental features to predict penetration bias values."**

**I would suggest removing "values" for consistency in the manuscript.**

Thank you for your observation. We have corrected it on page no.6, line no.201.

**4. Figure 1. The figure seems unnaturally flattened. Moreover, the map of Antarctica should typically be in polar stereographic projection visualized with an equal aspect ratio, so I would not recommend flattening it.**

**I wonder what the point of showing the IMBIE drainage basins is...the concept of drainage basins is not used in the rest of the manuscript.**
**Caption: "...for which ICESat-2 data, TanDEM-X data and ECMWF features is extracted" -> "...features are..."**

Thank you for your helpful suggestions. We ensured that the map projection used is EPSG 3031 (Antarctica polar stereographic) and added it in the figure caption. Additionally, we have corrected the image dimensions to ensure an equal aspect ratio for improved clarity. We have deleted the IMBIE drainage basins as requested.

**5. Lines 99–104. Salinity is mentioned here multiple times, but is it useful for the application over the Antarctic Ice Sheet?**

Thank you for your helpful suggestions. We have included salinity in the introduction solely to discuss the factors that affect X-band penetration depth. However, we agree that salinity has limited relevance for inland Antarctic regions, so we have added the sentence to specify that it primarily pertains to marine ice and coastal regions on page no.3, line no.92-93 as given below:

As such, X-band is more sensitive to changes in snow salinity in comparison with other microwave bands in the marine ice and coastal regions (Nandan et al., 2017).

**6. Line 106. "InSAR coherence also effects the penetration depth..."**

**This sentence is grammatically and logically wrong. I would recommend writing "a deeper penetration depth results in a lower InSAR coherence due to an increase in volume scattering (Deng et al., 2024)" or something similar.**

Agreed. The sentence is corrected on page no.3, line no. 98-99 as given below:

Penetration depth and InSAR coherence are closely related, as deeper radar penetration leads to lower InSAR coherence due to stronger volume scattering effects (Deng et al., 2024).

**7. Line 111. I am not sure that the cited work says "overall backscatter intensity decreases due to surface scattering". What I found in Park et al. (2021) is: "backscatter over wet snow**

**can also be increased due to the complex wet snow metamorphism, including an increase in snow surface roughness and an increase in the snow grain size during overnight refreezing".**

We have added and clarified this explanation in the revised manuscript on page nos.3-4, line nos.103-106, as given below:

As a result, backscatter is primarily from the air-snow interface, and overall backscatter intensity decreases due to volume scattering ( Park et al., 2021). However, during the ripening stage, backscatter over wet snow can sometimes increase because of processes such as surface roughening and grain growth associated with refreezing.

Park et al.(2021) demonstrate that in wet snow regions, backscatter is primarily from the air-snow interface, and further in the ripening stage, backscatter can also be increased due to surface roughening and grain growth.

**8. Lines 114–117. It is nice to distinguish the different concepts between "penetration depth" and "penetration bias", but it would be nicer to clarify again (apart from the title) that this study focuses on the bias.**

Thank you for your helpful suggestion. This is now corrected on page no.4, line no.117 as given below:

The primary focus of this study is on estimating penetration bias using TDX and ICESat-2 elevation data.

**9. Lines 126–131. Is there any reference for the concepts mentioned here? It would also be more concise and clearer if the relationships could be expressed in the format of equations. For example, it seems that Eq. 9 of Dall (2007) is helpful.**

Thank you for your comment. We have added the reference on page no.4, line no.123 as (Page & Ramseier, 1975), and equations (1), (2) are also added on page no.4, line no.129-135, for better explanation as given below:

This bound on elevation bias depends on both coherence ($\gamma$) and the height of ambiguity ($h_a$), which is the height difference that generates an interferometric phase change of $2\pi$. Equations (1) and (2) describe this bound:
$$\gamma = \frac{1}{1+j2\pi\delta_2/h_a} \tag{1}$$
$$\text{if and only if} \quad \Delta h =< \gamma h_a/2\pi \tag{2}$$
In regions of low coherence, the penetration depth and bias tend to be higher, whereas in high coherence regions show less penetration depth and minimal bias values.

**10. Line 134. It would be nicer to briefly explain to the readers of The Cryosphere what "interferogram flattening" is.**
Interferogram flattening is defined as the removal of the flat-Earth phase component caused by satellite imaging geometry. The explanation of interferogram flattening is added on page nos.4-5, line nos.137-140 as given below:

Interferogram flattening is defined as the removal of the flat-Earth phase component caused by satellite imaging geometry, and is a fundamental step to isolate phase signals due to topography or surface displacement. This gives an accurate measurement of surface deformation or elevation changes, as it removes the phase component caused by Earth's topography.

**11. Line 146. The penetration depths between 10 m and 12 m seem contradictory to what is mentioned below. These values also seem to deviate from Table 6 of Rizzoli et al. (2017).**

Rizzoli et al.(2017) reported a two-way penetration depth of 5.5 m over Greenland (Figure 15 (a)), derived using a physical model that likely underestimates the actual values. Whereas Figure 15(b) shows that the mean differences between TanDEM-X DEM and ICESat measurements reach up to 8.1m. In Table 6 of their study, they have compared the mean two-way penetration depth and penetration bias across all four facies regions of Figure 9 (a). Given the colder and drier conditions in Antarctica, similar or even greater penetration biases are plausible, with X-band radar potentially penetrating up to 10-12m in certain regions. We have confirmed these values also with Paola Rizzoli, who is a coauthor in this study. To reflect this change, we have added a reference mentioning (Rizzoli 2025 Personal Communication)

**12. Line 152. The Michel et al. (2014) work is not about InSAR penetration bias, but radar altimeter. Suggested removing.**

Following your recommendation, we have removed the above-mentioned reference on page no.5, line no.157.

**13. Line 167. Please specify which "previous models" were applied for which purposes, and please also provide references.**

The paragraph in question is a direct continuation of the preceding discussion, summarizing the prior models and references already mentioned to provide context for our study, as given below:

In previous studies, penetration bias has been evaluated by various data-driven models by interpreting the scattering mechanism, which depends on the physical properties of surface snow (Hoen & Zebker, 2000; Stebler et al., 2005; Dall, 2007; Oveisgharan & Zebker, 2007; Abdullahi et al., 2019; Fischer et al., 2019; Park et al., 2021). Models such as the coherent backscatter method (Dall, 2007; Abdullahi et al., 2019; Fischer et al., 2020) are calibrated by applying penetration biases inferred through bistatic coherence loss. Further penetration biases are also estimated by conducting an empirical analysis of snow facies zones, as these features reflect specific scattering attributes of the snow in that region and hence can be utilized for the approximation of bias (Fahnestock et al., 1993; Rignot et al., 2001; Falk et al., 2015; Gray et al., 2015). Additionally, fuzzy classification as an effective approach is also applied to demarcate facies for further bias estimation (Rizzoli et al., 2017b). All the previous modelling approaches estimated the penetration bias by relating to a few variables, such as coherence, SAR backscatter intensity with underlying assumptions, or by equating some environmental parameters, such as snow water equivalent, snow density, snow liquid water content, and top layer snow temperature to the backscatter intensity (Park et al., 2021).

However, we have also clarified it on page no.6, line no.174-175 as given below:

The Prior models discussed above were applied either to image datasets or to a small region where an altimetry dataset was available for bias calculation.

**14. Line 182. "These data have been collected between..." could be misunderstood, possibly indicating that ICESat-2 collects data only in winter seasons. I would recommend the authors be specific that they are the ones who selected these winter data.**

Thank you for your helpful suggestion. Following your recommendation, we have clearly mentioned it on page nos.7-8, line no. 220 to 222 as given below:

For this study, we specifically selected ICESat-2 data acquired between April and June, which corresponds to the typical winter season, during the years 2021 to 2024.

**15. Lines 189–194. I understand the authors would like to have a detailed motivation and background introduction about the time difference between TanDEM-X and ICESat-2. However, adding too much information before introducing the time difference adopted by the authors dilutes the real message. I would recommend that here it is sufficient to only keep Lines 195–199.**

Thank you for your valuable suggestion. We have corrected the sentences on page no.8, line no.227-231:

However, in high-accumulation or dynamic zones (e.g., coastal Antarctica or glaciers), shorter time gaps, ideally less than a year, are preferred to minimize vertical discrepancies due to surface evolution. For these reasons, we select a maximum time difference between TDX and ICESat-2 acquisitions of less than or equal to 30 days. Further, 14 environmental features are also added from ECMWF ERA-5, available on an hourly basis, at a horizontal resolution of 0.25 arc degrees are added for all these data points. Details of all these features are given in Table 1.

**16.Line 209. "To retrieve elevation profiles, the collected dataset of point elevation measurements is sampled by utilizing different beam spacings and arbitrary repeat-track geometries, along with the estimation of Root Mean Square (RMS) errors result into the elevation change measurements."**

**The sentence does not seem optimal, both grammatically and logically. From what I understood, I would suggest rewriting into "The elevation profiles are derived by sampling the collected dataset of point elevations using different beam spacing and arbitrary repeat-track geometries. The elevation change measurements are derived by computing the root mean square errors (RMSE)", or something similar. I am also not sure if I understood the differences (or similarities) between "elevation profiles" and "elevation change measurements". Hopefully the authors could clarify this.**

**In addition, throughout Section 2.1, I found it difficult to distinguish which part is the characteristics of ICESat-2 ATL06 product, and which part is post-processing of the authors. I would suggest the authors split the two processes in two separate paragraphs.**

Thank you for your comment. We have corrected the sentence on page no. 10, line nos.257-258. Elevation profiles are the absolute surface elevations measured along a satellite track at a given time. Elevation change measurements are calculated by comparing elevation profiles acquired at different times over the same location, further estimating surface height changes. As recommended, we have made two paragraphs on page nos.10-11 as given below:

The ICESat-2, launched in 2018, was a follow-up mission of ICESat, which was operational from 2003 to 2009(Zwally et al., 2002). It consists of an Advanced Topographic Laser Altimeter System (ATLAS) that transmits a laser beam with a wavelength of 532nm at a pulse repetition frequency of 10kHz (Markus et al., 2017). ICESat-2 operates with three pairs of laser beams, with each pair separated by 3km in cross-track with a pair spacing of 90 m, which enables ICESat-2 to determine the local cross-track slope. The nominal diameter of the footprint of each of the beams is 17m with an along-track sampling interval of 0.7 m due to a high repetition rate of 10 kHz (Markus et al., 2017). For Land Ice measurement, the ATL06 dataset (Smith et al., 2019) is utilized, having a resolution of 20 m along track (using a +/- 20 window) and an elevation accuracy of better than 3cm (Brunt et al., 2019).
      The elevation profiles are derived by sampling the collected dataset of point elevations using different beam spacing and arbitrary repeat-track geometries. By comparing these profiles across different times and estimating the Root Mean Square (RMS) errors, elevation change measurements can be evaluated. For interpolating the beam track to the reference ground track (RGT), ICESat-2 requires controlling the beam position to less than half the pair separation. The orbit of ICESat-2 is inclined at 92° along with measurement range up to 88° north and south, with a repeat cycle of 91 days (Markus et al., 2017). The elevation precision of the data depends on three factors, namely signal-to-noise ratio, the distance at which laser footprints are collected, and the precision in timing the photons. Each beam pair consists of strong and weak beam lasers, which facilitates the calculation of cross-track slope, an important parameter in cryosphere research. The NASA cap toolkit (Paolo and Nilsson 2025) was used to produce and process the ICESat-2 data.

**17. Line 227. "TanDEM-X is also designed to maintain a sharp balance between height of ambiguity with phase disambiguation (Zink et al., 2021)."**

**I wonder how important this balance is, as it does not seem to be mentioned in the result analysis. If it is really so important, I also recommend the authors introduce what phase disambiguation refers to the readers of The Cryosphere.**

The term 'phase disambiguation' refers to phase unwrapping in this context. This sentence was intended solely to introduce the TanDEM-X data. As recommended, a brief explanation of phase disambiguation is added on page no.2, line no. 50-53 as given below:

Phase disambiguation, also known as phase unwrapping, is the process of resolving the modulo-$2\pi$ ambiguity integrated in interferometric phase measurements to retrieve absolute surface displacement or elevation differences from wrapped interferometric data

**18. Line 229–231. The data acquisition period is repetitive, as it was already mentioned in Line 182. Meanwhile, most of the acquisition periods in Table 2 are 2021 to 2023 instead of 2024. Please be consistent.**

The years of data acquisition are reiterated in section 2.2 in the context of the detailed description of the TanDEM-X datasets used in this work. As shown in Table 2, the Tottem glacier and Stancomb-Wills glacier regions include datasets from 2021 to 2024. Therefore, we have indicated the acquisition period as 2021 to 2024.

**19. Lines 232–240. Is it possible to add a flowchart to show how ITP and TAXI work?**

Thank you for your valuable suggestion. As recommended, the flowchart of ITP and TAXI is attached in the Annexure as Figures A5 and A6 in the revised manuscript.

**20. Line 249. "Fourteen atmospheric features complement our dataset (Table 1)."**

**Why would the authors use these fourteen features to "complement" the dataset? This introductory sentence is quite abrupt and does not really show the importance of using these features. Meanwhile, these features were called "environmental features" and now "atmospheric features". Please be consistent. Strictly speaking, snow density should also not be part of "atmospheric features".**

**As an additional comment on the Data section, I recommend clearly stating the purposes of each dataset used by the authors.**

The term "atmospheric features" is replaced by "environmental features" throughout the manuscript. As recommended, an explanation for selecting each environmental feature is added on page no.12, line no.292-302, as given below:

The selection of environmental variables in this study is based on the physical processes affecting X-band penetration into snow and firn. Snow density, snow depth, and temperature of the snow layer directly affect the dielectric properties and stacking of snowpack and hence attenuate the radar signal. Variables such as 2m air temperature, snowfall, total precipitation, and snow evaporation are related to snow accumulation and melt processes, which affect snow structure and moisture content. Surface pressure may indirectly affect radar signal propagation by air density variation. Wind speed and direction influence snow redistribution and surface roughness, which influence radar backscatter and penetration. Various types of albedo parameters, such as near-infrared, UV-visible, and snow albedo, are related to the physical properties of snow, which affect volume scattering. The total column of snow water equivalent is related to snow accumulation and density, affecting the penetration depth of the radar signal. All these environmental features, which capture the variation of penetration depth due to diversity in snow and ice surface, and thus included to improve the machine learning and Deep learning model prediction for penetration bias estimation.

The same explanation is also given in comment no.2.

**21. Line 259. I wonder what the added value of repeatedly mentioning (also the modifications made to) the CSV files throughout the Methodology section is. While I appreciate that the authors provided the code and introduced the correct data format to the potential users, I believe such specifications would be more appropriate in the code documentation. Given the already extensive length of the manuscript, I suggest omitting these minor details.**

Thank you for your suggestion. We moved these details to the GitHub documentation

**22. Figure 3. I would appreciate it if the input and output of the models could also be specified.**

Thank you for your suggestion. As recommended, Figure 3 is corrected on page no.13.

**23. Lines 270–272. My concerns are the same as comment 21. The projection is not critical compared to the core logic and methodology of this manuscript.**

Thank you for your suggestion. We have removed this sentence in the current version of the manuscript.

**24. Lines 279–282. "Without any correction applied the differences between ICESat-2 elevation data and TanDEM-X DEM height are characterized by an RMSE of 4 m and median and mean value of -1.6 and -1.48 m along with minimum, maximum values as -18 m, 20 m respectively in line with previous studies (Rizzoli, et al., 2017)."**

**This sentence here is unclear with a grammatical mistake. Suggested change: "Without any correction applied, the differences between ICESat-2 heights and TanDEM-X heights are characterized by an RMSE of 4 m, a median of -1.6 m, a mean of -1.48 m, a minimum of -18 m, and a maximum of 20 m. The statistics are consistent with Rizzoli et al. (2017)."**

**Please feel free to improve the idea based on this version. In general, I suggest the authors add commas between clauses and consider splitting long and detailed sentences to improve clarity.**

As recommended, the correction is done on page no.14, line nos.323-326 as given below:

Prior to applying any corrections, the differences between ICESat-2 height and TDX height are characterized by an RMSE of 4 m, with median and mean values of -1.6 and -1.48 m, respectively, a minimum of -18 m, and a maximum of 20 m. The statistics are consistent with previous studies (Rizzoli et al., 2017a).

**25. Line 283. "Firstly, we have removed the data points lying in the floating ice region by removing the data points having TanDEM-X DEM height less than 100m."**

**I am curious why the authors are not interested in the floating-ice region. Moreover, this sentence could be more concise. I would suggest revising into "Points with TanDEM-X DEM heights below 100 m were removed to exclude the floating-ice region."**

We have removed the data points lying in the floating ice region for several reasons. Over floating ice regions, X-band penetration is much reduced, so penetration bias may occur due to tidal displacement or unwrapping errors (for detached icebergs), rather than SAR penetration, and including those data points could lead to incorrect interpretation. As recommended, the correction is done on page no.14, line no.327.

**26. Line 286. What does "finite accuracy" mean? Why is it important for this study?**

Finite accuracy means that the estimated values of baseline, including both its parallel and perpendicular components, are subject to measurement noise, instrument limitations, or calculation errors. We have rephrased this sentence in the manuscript on page no.14, line nos.329-330, as given below:

Similarly to Gonzalez et al. (2024), we consider that InSAR DEMs have finite accuracy due to uncertainties in satellite baseline measurements.

**27. Line 287. "For heights of ambiguity ranging between 30 and 80 meters expected tilts in a scene range between 7 to 19 cm (Gonzalez et al., 2024)."**

**If I understood correctly, did the authors try to say "For heights of ambiguity ranging between 30 and 80 meters, the expected tilts in a scene range between 7 cm to 19 cm (Gonzalez et al., 2024)"? I am not sure if I understood the purpose of mentioning this range "7 cm to 19 cm". Is it somewhat connected to the previous sentence "low-varying offsets and tilts"? Does it work as a quality indicator? Is it useful for the setup of the following experiments? Please specify.**

To filter the input data based on the height of ambiguity, we referred to findings from previous studies. According to Gonzalez et al. (2024), for heights of ambiguity ranging between 30 and 80 m, the expected vertical tilts in DEMs range from 0.07 m to 0.19 m. These values are considered as a reference to determine for acceptable range of height of ambiguity for filtering out the data points that would exceed the title range mentioned in the previous studies. This ensures that the DEM height is unbiased by baseline-induced height errors. We have corrected this on page no. 14, line no.332-334.

**28. Line 289. If the points with a height of ambiguity less than 30 m only take up 1% of the dataset, how do they introduce imbalance? Should we consider them as outliers? What about the points with a height of ambiguity higher than 80 m? Why would the authors not apply the same outlier removal criteria shown in Eqs. 1 and 2 also for the height of ambiguity?**

Thank you for your questions. Regarding the imbalance effect, a height of ambiguity (HOA) lower than 30 m introduces imbalance in the dataset, as the network will not have enough samples in its training dataset to be able to correctly predict data with such HOA. This is a common paradigm in AI. We cannot technically consider them as outliers as their elevation is not less accurate compared to other DEMs.
We do not remove any data points with higher values of HOA, as none of the data points in our dataset exhibit a height of ambiguity greater than 80 m. Finally, please note that the interquartile

range (IQR) filter acts on different input variables. We conducted multiple trials to pre-filter the data based on SAR features. This approach was adopted to ensure that no relevant or essential data points were inadvertently excluded during the outlier removal process.

**29. Lines 319–323. Many of the abbreviations do not seem to match the ones in Figs. S1–S4. I suggest double-checking.**

Thank you for your observation. The abbreviations used in Figures S1-S4 are defined in the caption of Figure S1, as including the full names within the figures will affect the clarity and readability.

**30. Table 3. Could the authors explain how the hyperparameters listed in the table are chosen?**

The listed parameters were obtained following the optimization of each respective model. We have added this in the Table 3 caption on page no.16, as well as given below:

For machine learning models, numerical parameters, and for deep learning models, the number of neurons in hidden layers, batch normalization with dropout, L2 regularization, and the number of training samples were iteratively optimized to minimize RMSE and MAE, maximize R2 score and explained variance, as well as the lowest mean and standard deviation of errors.

**31. Line 335. I would avoid using words like "obviously", as they can sound somewhat casual, and the observation may not be as obvious to the reader as it is to the authors. This sentence has another problem. Please refer to item 11 of technical corrections.**

As recommended, correction is done on page no.18, line no.383-385 as given below:

The elevation values of both TDX and ICESat-2 are highly correlated. Various environmental features, such as ultraviolet albedo and near-infrared albedo, and snow depth, show high values of correlation.

**32. Line 339. "Based on this analysis we decide to remove some of the highly correlated variables and leave only seven features including temperature of the air at 2m above the land surface, snowfall, surface pressure, snow density, snow evaporation, wind speed and wind direction." -> "Based on this analysis, we removed several highly correlated variables and kept the following seven features: 2 m air temperature, snowfall, surface pressure, snow density, snow evaporation, wind speed, and wind direction."**

**But it seems like this selection is only applied to the so-called "environmental features". The following text gives the impression that the so-called "radar features" are all used. If this is the case, I suggest the authors state clearly what exactly the retained features are.**

We have corrected the sentences on page no.18, line nos.387-391 to more clearly specify the radar and environmental features used as input variables for the AI models, as given below:

Based on this analysis, we removed some of the highly correlated environmental features (Table 5 * variables) and kept the following seven features: temperature of the air at 2m above the land surface, snowfall, surface pressure, snow density, snow evaporation, wind speed, and wind direction. Removing the features that do not show much variation in the histograms (Figure 5) helps reduce redundancy, improve learning efficiency, and mitigate risks such as overfitting and optimization challenges (Halkjrer & Winther, 1996). However, for radar features, all six features are taken as input.

**33. Figure 6 caption. "ICESat-2 elevation data is not used as input; only TanDEM-X DEM values are considered for the AI models."**

**Please check the consistency of the manuscript, as "data" should generally be plural and seems to be also used that way in most parts of this manuscript. Regarding this sentence itself, I find it appearing rather randomly. It does not really describe the figure, and the authors filtered out a lot of other parameters apart from the ICESat-2 heights, so why would the ICESat-2 elevation data be mentioned here only?**

Thank you for your observation. We have removed this sentence from the Figure 6 caption.

**34. Line 361. I find it odd to suddenly mention passive sensors. They are not used in this study as far as I understood.**

Thank you for your observation. We have removed this sentence on page no.22, line no.428.

**35. Line 366. This sentence is unclear due to grammatical issues and dense structure. If I understood correctly, it would be better to rewrite it into "Initially, all models are trained using 6 radar features as input: TanDEM-X DEM height, coherence, amplitude, height of ambiguity, local incidence angle, and slope. Penetration bias is used as the target variable." Please also decide whether the authors prefer to use TanDEM-X or TDX throughout the manuscript.**

Thank you for your observation. We have rephrased this sentence on page no.22, line nos.433-434, and now use TDX in the entire manuscript.

**36. Line 368. This is another sentence that I am not sure how to interpret due to grammatical issues and dense structure. According to my understanding, I would suggest rewriting into "For a 30-day time difference, the dataset contains 276,549 data points as input. This number is reduced to 215,620 when considering a 15-day time difference." Please also check for consistency when you mention "time difference" or "time interval".**

Thank you for your observation. We have implemented this change at page no.22, line nos.434-435 as given below:

For a 30-day time difference, the dataset contains 276,549 data points as input. This number is reduced to 215,620 when considering a 15-day time difference.

**37. Line 383. "...RMSE values nearer..." I cannot see the point of using a comparative form of an adjective. Please rephrase.**

**It is also important to mention the units of the errors throughout the Results section.**

Thank you for your comment. We have implemented these changes on page no.23, line no.450-451 as given below:

Similarly, the Decision Tree is the least effective model among the machine learning models, which gives an RMSE of approximately 1.5 m and an MAE value of 1.08 m for both 30 days and 15 days' time differences (Figure A1).

**38.Line 384. "The reason may be cited..." is quite unconventional. Based on my understanding, here is a suggested revision: "The probable reason for the compromised performance of the Decision Tree regression model is that it uses a single tree, which makes decisions by splitting features at threshold values." Please feel free to improve based on this, if I understood it wrong.**

Thank you for your comment. We have reworded this sentence accordingly on page no. 23, line no.451-454 as given below:

The lower performance of the Decision Tree regression model may be explained by the model structure, which relies on a single tree for making predictions based on threshold splits of input features. As this model is simple, it often performs poorly when complex interactions exist among variables.

**39. Line 393. The introduction of the features is repetitive. There should not be a comma after "including" in this sentence.**

A comma is removed in the sentence on page no.23, line no.462. Line nos. 462 and 463 serve as the introductory statements for section 4.2.

**40. Line 395. This sentence is difficult to follow due to missing verbs, dense structure, and inconsistent pluralization. Please consider rewriting into "Table 5 shows that DNN is the best-performing model among all the data models with an RMSE of approximately 1.04 m, an MAE of 0.74 m, an R2 score of 0.84, and an explained variance of 0.84", or something similar. In general, please specify the reported metrics precisely and avoid vague phrases such as "almost equal to" or "along with high corresponding values".**

As recommended, this correction is done on page no.23, line nos.463-465 as given below:

As presented in Table 5, which shows the model prediction parameters for input data with a 30-day time difference (Test Case 3 in Table 4), the DNN emerges as the best-performing model among all data models, achieving an RMSE of approximately 1.04 m, an MAE of 0.74 m, an $R^2$ score of 0.84, and an explained variance of 0.84.

**41. Figs. 7–10. For a more straightforward comparison, I would suggest combining them into one figure. There could be four subplots showing four scatter-plots, and one subplot showing all histograms in different colors. In addition, these figures, together with Fig. 4, seem unnaturally stretched.**

Thank you, we have now merged these figures and mentioned them as figure no.7 on page no.25.

**42. Line 401. "Similarly, Table 5..." Please refer to comment 40 to improve clarity.**

This is corrected on page no.24, line no.470 as given below:

Similarly, as presented in Table 5,..

**43. Line 405. "Figure 9..." The sentence has the same problem as the comments mentioned above. Suggested rewriting: "Figure 9 shows the DNN model results using data with a 15-day time difference. Compared to the case with a 30-day time difference (Figure 7), a slight improvement in performance is observed." Please also check the consistency of the preferred word for "time difference between TanDEM-X and ICESat-2" throughout the manuscript.**

We have rewritten this sentence as suggested on page no.24, line no.473-474 as given below:

Figure 7(C) shows the DNN model results using data with a 15-day time difference.

**44. Line 407. "...reduced between" is incorrect. Please specify: is it "reduced to" or "reduced by"?**

Thank you, we specify this "reduced by" and it is corrected on page no. 24, line nos.475-477 as given below:

Overall, including ECMWF environmental features as inputs to the neural network algorithms resulted in RMSE and MAE being reduced by 10 to 20 cm, while both the R2 score and explained variance values showed improvement.

**45. Lines 410–419. This is a rather extensive description of the Supplementary material. As already mentioned in general comments, would it be clearer to put everything in Appendix?**

Thank you for your suggestion. As recommended, Figures A1, A2, A3, and A4 are added in Appendix A. The required corrections are done on page no.22, line no.441; page no.23, line nos.451,456, and page no.42.

**46. Line 435. "As the DNN model with model architecture given in Figure 11 is most efficient...a SHAP analysis..." This sentence seems to imply that SHAP analysis was performed only because the DNN model showed the best performance. However, Line 459 mentions that SHAP analysis was also conducted for the RF model and the corresponding figures are not shown. This gives the impression that SHAP was applied to both models regardless of performance. To avoid confusion and to improve the flow, I recommend briefly**

**introducing the SHAP analysis and its purpose in the Methods section, so that its use in the Results section does not seem unexplained.**

Thank you for your recommendation. We have clarified this section related to the SHAP analysis. The new section, no.3.4, is added on page no.21 as an introduction to SHAP analysis.

**47. Line 452. Would the result change if the authors use more or less than 1000 samples in the SHAP analysis?**

Thank you for your question. Increasing or decreasing the sample size may vary the variance in the estimates, but it is not expected to alter the primary feature importance rankings, thanks to the large dataset size having more than 200,000 datapoints. In our analysis, we repeated the experiments multiple times to ensure consistency, and the results remained stable across runs.

**48. Line 457. "...wind speed, wind direction and surface pressure which also shows broad data distribution..." Did the authors mean surface pressure shows a broad data distribution? But from what I understood, all input features should show a broad distribution. What is the purpose of mentioning it here?**

Thank you, we removed this statement on page no.26, line no.491-492 in the updated version of the manuscript.

**49. Line 475. When the authors mentioned the corrected TanDEM-X DEM, is it possible to provide a map indicating the difference before and after the correction? I assume it can be interesting to see the spatial patterns of the improvement, apart from pure statistics.**

Thank you for your valuable suggestion. We have selected a small subset of DEM to demonstrate the effect of the proposed method. The figure of DEM showing before and after correction is shown in Figure 2.

[Figure]

**Figure 2. Comparison of TanDEM-X original DEM with TanDEM-X bias corrected DEM (i) Map of Antarctica showing the four selected locations (A-D), (ii) DEM comparison for each location is shown in corresponding rows. The TanDEM-X bias-corrected DEM raster is calculated by subtracting the penetration bias predicted by the DNN model trained on 15 days' time difference data from the TanDEM-X original DEM. Penetration bias and histogram for each location are also plotted.**

**50.Line 481. This was already mentioned in general comments. When the authors mentioned that the corrected TanDEM-X DEM achieves an accuracy similar to that of stereophotogrammetric DEMs, it would be interesting to see a comparison with REMA.**

Thank you for your suggestion. Please refer to comment no.5 in the general comments. Comparisons of TanDEM-X DEMs with the REMA DEM are shown in Figure 1.

**51. Line 482. It would be clearer to specify that the "block-wise calibration" is a procedure used in the Wessel et al. (2021) study. I am also not sure if I understood "residual elevation scatter". According to the Wessel et al. (2021) study, "For the HPB the mean penetration bias and standard deviations vary from −1.68 to −5.66 m and 0.92 to 1.20 m, respectively. They are located in the interior of Antarctica and are well distributed over the continent to serve as ground control for the adjacent blocks." These values (0.9 m to 1.2 m) seem to refer to the standard deviation of penetration bias.**

Thank you, we agree. We have corrected this sentence on page no.29, line no.516 as given below:

the standard deviation of the penetration bias was ~0.9–1.2 m (Wessel et al., 2021).

**52. Line 486. "For example...DEM products can be integrated with other datasets..." This sentence is extremely vague. Please specify how to integrated the DEM with which datasets for which purposes.**

We have rephrased this sentence now referring to DEMs on page no.29, line no.520-523 as given below:

For example, surface elevation changes, flow dynamics, and grounding line positions derived from the DEM will be more reliable. Moreover, mosaic DEM products (such as the TDX PolarDEM) can be integrated with other DEMs without requiring large bias corrections (Kim & Kim, 2017; Milillo et al., 2019; Milillo et al., 2022).

**53.Line 490. Change detection is a similar idea to the "elevation change" concept mentioned in the following section. Would the authors consider merging the two sections and making the story more concise?**

Agreed, we have removed this sentence on page no.29, in which "change detection" was earlier mentioned.

**54. Line 498. "This is critical because even modest systematic errors can translate into..." It would be better to specify that these errors are in heights.**

We agree. We modified the manuscript on page no.29, line 530.

**55. Line 509. "Notably, in radar percolation zones where seasonal melt occurs, uncorrected InSAR measurements can differ by up to 10–15 m from the actual surface (Wessel et al., 2021)."**

**I am not sure if I follow this logic. In principle, percolation zones with seasonal melt should be characterized by high-density rough layers on the surface and near the surface. I would not expect such a high penetration bias. I looked up how Wessel et al. (2021) commented on the difference between dry and percolation zones: "Here, the temperatures are coldest (Macelloni et al., 2019; Scambos et al., 2018), and the SAR signal penetrates the most in dry, cold firn (Ulaby et al., 1986), whereas the coastal areas show lower penetration which clearly corresponds to the brighter reflecting percolation areas in the amplitude mosaic (Fig. 11). This variation in the SAR penetration over the whole AIS raises the absolute linear error (LE90) to 6.25 m, which is calculated by sorting the absolute differences thresholded by 90 % of the values." I strongly recommend the authors specifying what they meant.**

Agreed, the right reference for this is Piermattei et al 2024, which we now cite instead of Wessel et al. 2021.

Piermattei, L., Zemp, M., Sommer, C., Brun, F., Braun, M. H., Andreassen, L. M., Belart, J. M. C., Berthier, E., Bhattacharya, A., Boehm Vock, L., Bolch, T., Dehecq, A., Dussaillant, I., Falaschi, D., Florentine, C., Floricioiu, D., Ginzler, C., Guillet, G., Hugonnet, R., … Yang, R.: Observing glacier elevation changes from spaceborne optical and radar sensors - an inter-comparison experiment using ASTER and TanDEM-X data, The Cryosphere, 18(7), 3195–3230, https://doi.org/10.5194/tc-18-3195-2024,2024.

**56. Line 514. "bias-corrected bistatic InSAR data": I assume the authors meant that they corrected for the DEM rather than the data themselves?**

Thank you for your observation. We replaced "bistatic InSAR data" with "bistatic InSAR DEMs" on page no.30, line no.547, as we are applying our corrections to the height values and not to the InSAR phase directly.

**57. Lines 531–535. I am afraid I got lost in the logic of this paragraph. The paragraph seems to imply that over a long (30-day) time difference (again, please make it consistent and choose between difference, period and interval) between TanDEM-X and ICESat-2, both surface changes and volume changes can occur. But the authors seemed to try to explain why coherence outperforms amplitude, which means that the volume change should be more influential than surface changes?**

We have rephrased this sentence on page no.30, line no.563-567 to improve clarity:

"The SHAP analysis suggests that over a 30-day time difference, coherence slightly outperforms amplitude in predicting bias. This likely owes to the amplitude sensitivity to any surface changes that might occur over a longer period between LIDAR and InSAR observations. In bistatic acquisitions, coherence can be reduced by volume scattering from deep layers; over 30

days, additional events affecting the surface could occur from events such as snowfall or wind redistribution, making coherence outperform amplitude as a predicting indicator. "

**58. Line 546. "...underscoring that encapsulates a different aspect of the scattering physics..." I tried to help revise this, but this part of the sentence is hardly understandable.**

Agreed, we have deleted this sentence on page no. 31, line no.578 in the updated version of the manuscript.

**59. Line 587–589. I am not sure if I understood it. Did the authors mean that the ECMWF data are in a coarse spatial resolution, whereas the real-world climate effects are rather local hence require a high-resolution model?**

Correct, we are saying that with a 30 km ECMWF resolution, there might be environmental variables that change locally.

**60.Line 610. "...confirms that SAR can serve as a powerful tool for measuring ice topography..." Would it be more precise to say it is bistatic InSAR? Furthermore, on Line 615, the concept of "swath-based InSAR" is suddenly introduced. I would strongly recommend the authors be more consistent with the language they use to refer to TanDEM-X.**

We agree. We now specify A swath-based continuous SAR as opposed to a point-wide altimetry approach. Corrections are done on page no. 32, line no. 638,641 and page no. 33, line no. 645.

**61. Line 620. "deep drawdown at a glacier terminus": does it mean a steep topography or a pronounced ice loss?**

It means steep topography. We have modified the text to improve clarity accordingly on page no.33, line no. 649-650 as given below:

This limitation is acute in dynamic zones: for instance, a narrow, deep, and steep topographic change at a glacier terminus…

**62. Line 630. I wonder why optical sensors should be mentioned suddenly.**

We mention optical sensors because we think that it puts in perspective the advantages of SAR vs other techniques. LIDAR is, per-se, an active coherence optical sensor, depending on the wavelength used.

**Technical corrections**

**1. Please check in Table 1 and throughout the manuscript the misspelling of ECMWF.**

Thank you. This is corrected in Table 1 on page no.9-10. This is also corrected on page no.32, line no. 614, page no.45, line no.837.

**2. Line 133. "...and interferometric phase change..." -> "...an..."**

This is corrected on page no.4, line no.130.

**3. Line 134 and throughout the manuscript. Please check that there should be a space between the text and the citation. Some commas and dots are missed here and there.**

Thank you. This is corrected throughout the manuscript.

**4. Line 135. "almost equals" -> "almost equal"**

This is corrected on page no.5, line no.141.

**5. There are two Rizzoli et al. (2017) reference entries that should be distinguished with 2017a and 2017b.**

This is corrected on page no.2, line no. 54; page no.5, line nos.151,170; page no.11, line no.282; page no.14, line no.325; page no.34, line no.690.

**6. Line 164 and throughout the manuscript. The use of "like" to introduce examples is informal in academic writing. It is strongly recommended replacing them with "such as".**

This is corrected on page no.1, line no.23; page no.4, line no.121; page no.5, line nos.165; page no.6, line nos.171,172,175,176.

**7. Line 197 and throughout the manuscript. The abbreviation TDX is used in parallel with TanDEM-X. I would suggest the authors remain consistent with only one expression.**

Thank you for your suggestion. Now, in the revised manuscript, the term "TDX" is mentioned for consistency.

**8. Line 205. "...with each pair is separated by distance of 3km..." -> "...with each pair separated by 3 km..."**

This is corrected on page no.10, line no.252.

**9. Line 207 and many other places in the manuscript. (B. Smith et al., 2019) -> (Smith et al., 2019)**

This is corrected on page no.10, line no.254.

**10. Line 214. "signal-to-noise ratio"**

This is corrected on page no.11, line no.262.

**11. Line 215 and throughout the manuscript. The present continuous tense is generally not considered appropriate in academic writing. The accepted tenses should be past simple, present simple and present perfect. It is strongly recommended thoroughly checking it throughout the manuscript.**

Thank you for your comment. This is corrected on page no.11, line no.263, and checked it in the entire manuscript.

**12. Line 273. "...the corresponding TanDEM-X elevation data and associated radar features is extracted from..." -> "...are extracted from..."**

This is corrected on page no.14, line no.316.

**13. Line 284. "We consider only points where the TanDEM-X coherence is larger than 0.3 as high coherence values lead to a more accurate estimate of the interferometric phase." -> "We consider only points where the TanDEM-X coherence is larger than 0.3, as high coherence values lead to a more accurate estimate of the interferometric phase."**

This is corrected on page no.14, line no.327-329.

**14. Line 285 and throughout the manuscript. When a reference entry is mentioned within the texts, it should be cited as "Gonzalez et al. (2024)". I recommend the authors double check the citation rules.**

**Also, this line, it should be "Similarly to Gonzalez et al. (2024), we..." For similar problems, please refer to item 24 of specific comments.**

This is corrected on page no.14, line no.329.

**15. Line 311. "Before data feeding in the machine learning and deep learning algorithms, data is divided into training, validation and testing data." -> "Before being fed into the machine learning and deep learning algorithms, the data is divided into training, validation, and testing sets," or "Before feeding data into the machine learning and deep learning algorithms, we divide the data into training, validation, and testing sets."**

This is corrected on page no.15, line no.357-358.

**16. Line 323. "The accuracy of all these machine learning and neural network models are compared..." -> "...is compared..."**

This is corrected on page no.15, line no.370.

**17.Line 334. "To understand potential interrelations between radar and atmospheric features we focus to plot histograms of our variables and calculate their respective correlation (Figure 5, Figure 6)." -> "To understand potential interrelations between radar**

and atmospheric features, we focus on plotting histograms of our variables and calculating their respective correlation (Figs. 5–6).”

This is corrected on page no.18, line no.381-382.

**18. Line 336. “Various environmental features like Ultraviolet Albedo and Near Infrared Albedo along with snow depth show high values of correlation.” -> “Various environmental features, such as ultraviolet albedo, near-infrared albedo, and snow depth, show high values of correlation.”**

This is corrected on page no.18, line no.384-385.

**19. Line 337 and multiple places in the manuscript. This is an incorrect use of “whereas”; it is not typically used to start a sentence.**

This is corrected on page no.11, line no.281; page no.18, line no.385; page no.23, line no.454.

**20. Line 344. “...ICESat-2 elevation data is excluded” -> “...data are...” Please also check throughout the manuscript for consistency.**

The same is corrected and checked in the manuscript. This is corrected on page no.18, line no.394.

**21. Line 376. “best performed” -> “best-performing”**

 This is corrected on page no.22, line no.442.

**22. Line 386. “However, it’s not much effective when there is a complex interaction between the features of the input data. Whereas Random Forest which is an ensemble of multiple decision trees is very effective in complex non-linear regression tasks.”**

**This paragraph contains serious issues in both grammar and clarity. The first sentence is grammatically incorrect (“it’s not much effective” is not standard English and should be avoided in academic writing). The second sentence should not start with “whereas”, and the contrast introduced by “whereas” is misused (it is difficult to understand what kind of comparison is achieved here). I strongly recommend rewriting this part to improve both clarity and readability.**

Thank you for your comment. This correction is done on page no. 23, line nos. 453-455.

**23.Line 399. “...a gaussian distribution of errors suggesting that...” -> “a Gaussian distribution of errors, suggesting that...”**

This is corrected on page no.24, line no.467.

**24. Line 404. A comma should be added after "Interestingly".**

This is corrected on page no.24, line no.472.

**25. Line 454. Suggested revision: "The top five features which influence the DNN model prediction ability include coherence, amplitude, height of ambiguity, TanDEM-X DEM height values, and 2 m temperature. These features are ranked by importance."**

This is corrected on page no.26, line nos. 488-490.

**26. Line 456. "effecting" -> "affecting" or "influential".**

This is corrected on page no.26, line no. 491.

**27. Line 456. "Other significant atmospheric features which are affecting the DNN model includes wind speed..." -> "Other significant atmospheric features affecting the DNN model include wind speed..."**

**In this case, one can use a present participle ("affecting"), as long as it serves as part of a reduced relative clause.**

This is corrected on page no.26, line no.490-491.

**28. Line 457. Suggested revision: "For 30-day-difference data, the top five features..."**

This is corrected on page no.26, line no. 491-492.

**29. Line 458. "The top performing features is coherence followed by amplitude. " -> "The best-performing features are coherence and amplitude."**

This is corrected on page no.26, line no. 492-493.

**30. Line 459. "...both types of datasets"**

This is corrected on page no.26, line no.493.

**31. Line 459. Suggested revision: "We also conducted a SHAP analysis for the Random Forest model, the second-best-performing regression model after DNN. The SHAP analysis of the Random Forest model gave the same result as the DNN model for both the 15- and 30-day cases."**

This is corrected on page no.26, line nos. 493-495.

**32. Line 475. "...height achieves..."**

This is corrected on page no. 29, line no. 509.

**33. Line 477. "about" is rather informal. I would change it into "approximately".**

This is corrected on page no. 29, line no. 511.

**34. Line 547. "...coherence excels when capturing volume scattering, while amplitude captures persistent surface characteristics, they are both essential." -> "...coherence excels when capturing volume scattering, while amplitude captures persistent surface characteristics; they are both essential."**

This is corrected on page no. 31, line nos.578-579.

[revised manuscript text omitted]

Kuipers Munneke, P., Luckman, A. J., Bevan, S. L., Smeets, C. J. P. P., Gilbert, E., van den Broeke, M. R., Wang, W., Zender, C., Hubbard, B., Ashmore, D., Orr, A., and King, J. C.: Intense Winter Surface Melt on an Antarctic Ice Shelf, Geophys. Res. Lett., 45, 7615–7623, https://doi.org/10.1029/2018gl077899, 2018.

Li, C., Jiang, L., Liu, L., & Wang, H. : Regional and Altitude-Dependent Estimate of the SRTM C/X-Band Radar Penetration Difference on High Mountain Asia Glaciers, IEEE Journal of Selected Topics in Applied Earth Observations and Remote Sensing, 14, 4244–4253, https://doi.org/10.1109/JSTARS.2021.3070362,2021.

Markus, T., Neumann, T., Martino, A., Abdalati, W., Brunt, K., Csatho, B., Farrell, S., Fricker, H., Gardner, A., Harding, D., Jasinski, M., Kwok, R., Magruder, L., Lubin, D., Luthcke, S., Morison, J., Nelson, R., Neuenschwander, A., Palm, S., … Zwally, J.: The Ice, Cloud, and land Elevation Satellite-2 (ICESat-2): Science requirements, concept, and implementation

Remote Sensing of Environment, 190, 260–273, https://doi.org/10.1016/j.rse.2016.12.029,2017.

Michel, A., Flament, T., and Rémy, F.: Study of the Penetration Bias of ENVISAT Altimeter Observations over Antarctica in Comparison to ICESat Observations, Remote Sensing, 6, 9412–9434, https://doi.org/10.3390/rs6109412, 2014.

Milillo, P., Rignot, E., Rizzoli, P., Scheuchl, B., Mouginot, J., Bueso-Bello, J. L., Prats-Iraola, P., & Dini, L. :Rapid glacier retreat rates observed in West Antarctica. Nature Geoscience, 15(1), 48–53, https://doi.org/10.1038/s41561-021-00877-z,2022.

Milillo, P., Rignot, E., Rizzoli, P., Scheuchl, B., Mouginot, J., Bueso-Bello, J., & Prats-Iraola, P. :Heterogeneous retreat and ice melt of Thwaites Glacier, West Antarctica, Science Advances,2019.

Nandan, V., Geldsetzer, T., Mahmud, M., Yackel, J., & Ramjan, S. : Ku-, X- and C-Band Microwave Backscatter Indices from Saline Snow Covers on Arctic First-Year Sea Ice, Remote Sensing, 9(7),https://doi.org/10.3390/rs9070757,2017.

Oveisgharan, S., & Zebker, H. A.:Estimating snow accumulation from InSAR correlation observations, IEEE Transactions on Geoscience and Remote Sensing, 45(1), 10–20, https://doi.org/10.1109/TGRS.2006.886196,2007.

Page, D. F., & Ramseier, R. O. :Application of Radar Techniques to Ice and Snow Studies, Journal of Glaciology, 15(73), 171–191, https://doi.org/10.3189/s0022143000034365,1975.

Park, J., Forman, B. A., & Lievens, H. : Prediction of Active Microwave Backscatter over Snow-Covered Terrain across Western Colorado Using a Land Surface Model and Support Vector Machine Regression, IEEE Journal of Selected Topics in Applied Earth Observations and Remote Sensing, 14, 2403–2417, https://doi.org/10.1109/JSTARS.2021.3053945,2021.

Rignot, E., Echelmeyer, K., & Krabill, W. :Penetration depth of interferometric synthetic-aperture radar signals in snow and ice, Geophysical Research Letters, 28(18), 3501–3504, https://doi.org/10.1029/2000GL012484,2001.

Rizzoli, P., Martone, M., Rott, H., & Moreira, A. : Characterization of snow facies on the Greenland ice sheet observed by TanDEM-X interferometric SAR data, Remote Sensing, 9(4), https://doi.org/10.3390/rs9040315,2017.

Rott, H., Scheiblauer, S., Wuite, J., Krieger, L., Floricioiu, D., Rizzoli, P., Libert, L., & Nagler, T. :Penetration of interferometric radar signals in Antarctic snow, The Cryosphere, 15(9), 4399–4419, https://doi.org/10.5194/tc-15-4399-2021,2021.

Slater, T., Shepherd, A., McMillan, M., Muir, A., Gilbert, L., Hogg, A. E., Konrad, H., and Parrinello, T.: A new digital elevation model of Antarctica derived from CryoSat-2 altimetry, The Cryosphere, 12, 1551–1562, https://doi.org/10.5194/tc-12-1551-2018, 2018.

Smith, B. E., Gardner, A., Schneider, A., and Flanner, M.: Modeling biases in laser-altimetry measurements caused by scattering of green light in snow, Remote Sensing of Environment, 215, 398–410, https://doi.org/10.1016/j.rse.2018.06.012, 2018.

Smith, B., Fricker, H. A., Holschuh, N., Gardner, A. S., Adusumilli, S., Brunt, K. M., Csatho, B., Harbeck, K., Huth, A., Neumann, T., Nilsson, J., & Siegfried, M. R.: Land ice height-retrieval algorithm for NASA's ICESat-2 photon-counting laser altimeter, Remote Sensing of Environment, 233, https://doi.org/10.1016/j.rse.2019.111352,2019.

Stebler, O., Schwerzmann, A., Lüthi, M., Meier, E., & Nüesch, D.: Pol-InSAR observations from an alpine glacier in the cold infiltration zone at L- and P-band, IEEE Geoscience and Remote Sensing Letters, 2(3), 357–361, https://doi.org/10.1109/LGRS.2005.851739,2005.

Tsang, L., Durand, M., Derksen, C., Barros, A. P., Kang, D. H., Lievens, H., Marshall, H. P., Zhu, J., Johnson, J., King, J., Lemmetyinen, J., Sandells, M., Rutter, N., Siqueira, P., Nolin, A., Osmanoglu, B., Vuyovich, C., Kim, E., Taylor, D., … Xu, X.: Review article: Global monitoring of snow water equivalent using high-frequency radar remote sensing, The Cryosphere,Vol. 16, Issue 9, 3531–3573, Copernicus Publications. https://doi.org/10.5194/tc-16-3531-2022,2022.

Warren, S. G.:Optical properties of ice and snow. In Philosophical Transactions of the Royal Society A: Mathematical, Physical and Engineering Sciences, Vol. 377, Issue 2146,https://doi.org/10.1098/rsta.2018.0161,2019.

Wessel, B., Huber, M., Wohlfart, C., Bertram, A., Osterkamp, N., Marschalk, U., Gruber, A., Reub, F., Abdullahi, S., Georg, I., & Roth, A. : TanDEM-X PolarDEM 90m of Antarctica: Generation and error characterization, The Cryosphere, 15(11), 5241–5260, https://doi.org/10.5194/tc-15-5241-2021,2021.

Zwally, H. J., Schutz, B., Abdalati, W., Abshire, J., Bentley, C., Brenner, A., Bufton, J., Dezio, J., Hancock, D., Harding, D., Herring, T., Minster, B., Quinn, K., Palm, S., Spinhirne, J., & Thomas, R. :ICESat's laser measurements of polar ice, atmosphere, ocean, and land, In Journal of Geodynamics ,Vol. 34, www.elsevier.com/locate/jog,2002.